# SALT: Structure-Aligned Learning for Time-Series Forecasting

## Abstract

Time-series forecasting is vital in domains such as economics, energy, and traffic. Although many models exploit the decomposable nature of time-series, they are typically trained with a single objective (e.g., MSE), which imposes structural limits on their performance. We empirically demonstrate that this paradigm gives rise to two characteristic challenges: gradient interference, where heterogeneous components conflict, and spectral bias, where dominant low-frequency structures overshadow informative high-frequency ones. To move beyond these limitations, we introduce SALT (**S**tructure-**A**ligned **L**earning for **T**ime-Series), which combines Iterative Dominant Extraction (IDE) with Separable Training to optimize components independently. Our theoretical and empirical analyses show that this regime reduces cross-term errors, balances convergence across frequencies, and consistently surpasses the conventional methods across six backbones and nine benchmarks.

## 1 Introduction

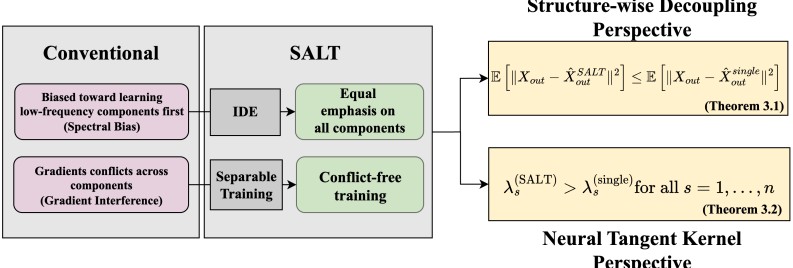

Figure 1: Conventional models trained with a single loss suffer from spectral bias (favoring low-frequency structures) and gradient interference (conflicting updates across components). SALT mitigates these issues by separating the optimization of decomposed components, yielding balanced learning across structures and theoretical error guarantees.

Time-series data, widely collected across domains such as energy, finance, climate, and healthcare, are inherently structured, exhibiting long-term trends, periodic seasonalities, and irregular fluctuations (Shumway et al., 2000; Hyndman & Athanasopoulos, 2018). These structures must be properly leveraged to ensure accurate and robust forecasting. Recent models have sought to incorporate such structures through decomposition-based designs (Wu et al., 2021; Zhou et al., 2022; Zeng et al., 2023), yet most continue to rely on a single-loss objective that jointly optimizes all components. This leads to two optimization challenges. First, gradient interference arises when structurally distinct components (e.g., smooth trends versus residual fluctuations) generate conflicting update directions, slowing or destabilizing convergence (Chai et al., 2022). Second, neural networks exhibit spectral bias, favoring low-frequency signals over high-frequency ones (Rahaman et al., 2019; Jacot et al., 2018). While this can help suppress noise, it also delays the learning of informative high-frequency structures, creating an imbalance across components. We empirically observe both effects. Figure 3 (Top) shows that gradients between dominant and non-dominant components frequently align negatively, evidencing interference under single-loss training. Figure 3 (Bottom) shows that low-frequency structures converge much faster than residual components, illustrating spectral bias in

real-world data. As a result, conventional models tend to overemphasize dominant patterns and underutilize finer-grained signals.

These challenges reduce stability and generalization, especially under non-stationary dynamics. In fact, recent time-series studies also highlight that gradient interference and spectral bias hinder effective learning (Feng et al., 2024; Fan et al., 2021; Das et al., 2020). Similar difficulties have been reported in related work, such as vanishing gradients in recurrent models (Bengio et al., 1994) and information bottlenecks in transformer-based designs when structural diversity is underexploited (Zeng et al., 2023).

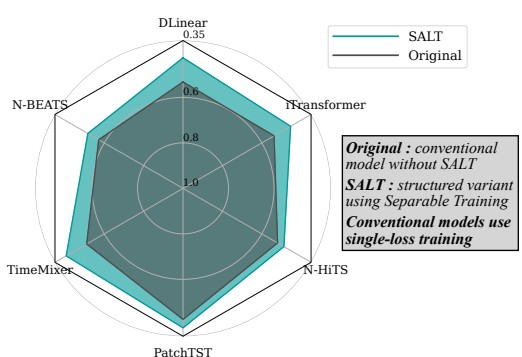

Figure 2: Performance comparison of six forecasting models with and without Separable Training. Each score represents the averaged MSE across nine benchmarks. Consistent gains across diverse architectures support the claim that structure-aligned learning improves robustness.

In this paper, we revisit forecasting from an optimization perspective. Our central claim is not a new architecture, but a theoretical and empirical case for training decomposed components separately. We study this regime under the name **SALT** (**S**tructure-**A**ligned **L**earning for **T**ime-Series). To illustrate the idea, we use lightweight decomposition procedures such as STL, STR, or frequency filters to obtain structural components, and then apply Separable Training so that each component is optimized independently. These choices are not the core contribution, but practical means of realizing the proposed learning regime.

We justify this approach with two complementary analyses. From a structure-wise error decomposition view, separating components reduces cross-term errors and stabilizes optimization. From an NTK-based spectral view, it flattens the eigenvalue spectrum, enabling more balanced convergence across components. Together, these analyses explain why Separable Training mitigates the limitations of single-loss optimization.

Finally, we evaluate this regime on six forecasting backbones and nine benchmark datasets. While the absolute gains are moderate, they are consistent across all settings, demonstrating that structure-aligned learning provides robustness and generality.

Our contributions are as follows:

1. We identify and empirically verify gradient interference and spectral bias as central optimization challenges in time-series forecasting.

2. We formalize structure-aligned learning as a principled alternative to single-loss optimization, emphasizing theoretical benefits of training decomposed components separately.

3. We provide complementary analyses — error decomposition and NTK-based eigenvalue spectrum analysis — that explain why this regime improves learning dynamics.

4. We demonstrate empirically that structure-aligned training yields consistent improvements across diverse architectures and datasets.

## 2 RELATED WORK

### 2.1 STRUCTURAL MODELING IN TIME-SERIES FORECASTING

Time-series data often exhibit structural properties such as long-term trends, seasonal cycles, and residual fluctuations. Recent forecasting models have sought to leverage these structures within architectural designs. For example, Autoformer (Wu et al., 2021) and FEDformer (Zhou et al., 2022) introduce attention mechanisms that explicitly separate trend and seasonality, while PatchTST (Nie et al., 2022) exploits local windows and channel independence to capture periodicity. Similarly,

N-BEATS (Oreshkin et al., 2019) and N-HiTS (Challu et al., 2023) adopt recursive prediction and residual-based learning to progressively isolate predictable signals.

Despite these advances, most approaches still rely on a single global loss (e.g., MSE) over the entire series. This training scheme entangles heterogeneous components: smooth, low-frequency trends are optimized rapidly, while high-frequency structures remain underfitted (Rahaman et al., 2019; Jacot et al., 2018). Such an imbalance leads to optimization conflicts across components, resembling the gradient interference phenomena observed in multi-task learning (Chai et al., 2022). As a result, single-loss training can limit generalization in multi-component time series.

In contrast, our approach explicitly couples structural decomposition with the training objective. SALT decomposes a series into dominant components using IDE, which iteratively extracts dominant parts, and then applies Separable Training to optimize each component independently. This structure-aligned training reduces gradient conflicts and alleviates spectral bias, providing a principled bridge between decomposition and optimization.

## 2.2 Optimization Challenges in Single-Loss Forecasting

While recent models incorporate structural components into their architectures, most still optimize all patterns jointly under a single loss (e.g., MSE). This global objective does not respect the heterogeneous learning dynamics of decomposed components, leading to two characteristic challenges.

**Gradient interference.** When gradients from different components are aggregated into a single update, they often point in conflicting directions. This phenomenon, widely studied in multi-task learning (Chai et al., 2022), also arises in time-series forecasting, where interference among time-series data (Feng et al., 2024; Fan et al., 2021). Smooth trend components and noisy residual components can produce opposite update signals, slowing or destabilizing convergence. In our experiments (Figure 3, Top), pairwise cosine similarities between component-specific gradients show that negative alignments occur frequently, indicating interference between structural patterns during single-loss optimization.

**Spectral bias.** Neural networks are also known to exhibit a preference for low-frequency signals (Rahaman et al., 2019; Jacot et al., 2018). In forecasting, this bias implies that dominant low-frequency structures (e.g., trends or seasonalities) are learned much faster, while high-frequency components converge slowly even when they contain predictive information. This challenge has also been recognized in time-series analysis, where spectral methods reveal similar frequency imbalances (Das et al., 2020). We confirm this phenomenon in real-world dataset (Figure 3, Bottom): independent training on decomposed components shows that the dominant pattern achieves rapid loss reduction, while others lag behind, evidencing an imbalance in convergence.

Together, these findings reveal that single-loss optimization introduces both conflicting gradient directions and frequency imbalance, neither of which is addressed by architectural decomposition alone. This motivates a structure-aligned training regime where decomposed components are optimized independently, a perspective we develop in the following sections.

## 2.3 Theoretical Insights via NTK Analysis

The rapid growth of deep learning has raised the need to understand training dynamics in a principled way. In this context, the Neural Tangent Kernel (NTK) has emerged as a powerful tool for analyzing wide neural networks (Jacot et al., 2018). NTK models the output of a neural network as a linear function in function space and provides a theoretical foundation for understanding how neural networks prioritize different features during training.

Traditional kernel methods measure similarity in a high-dimensional feature space via a kernel function:

$$k(\mathbf{z}_i, \mathbf{z}_j) = \phi(\mathbf{z}_i)^\top \phi(\mathbf{z}_j), \tag{1}$$

where $\mathbf{z}$ denotes the input sample, and $\phi(\cdot)$ maps input $\mathbf{z}$ into a high-dimensional space. In the NTK setting, the feature mapping is defined via the model's gradients:

$$\phi(\mathbf{z}) = \nabla_\theta f_\theta(\mathbf{z}), \tag{2}$$

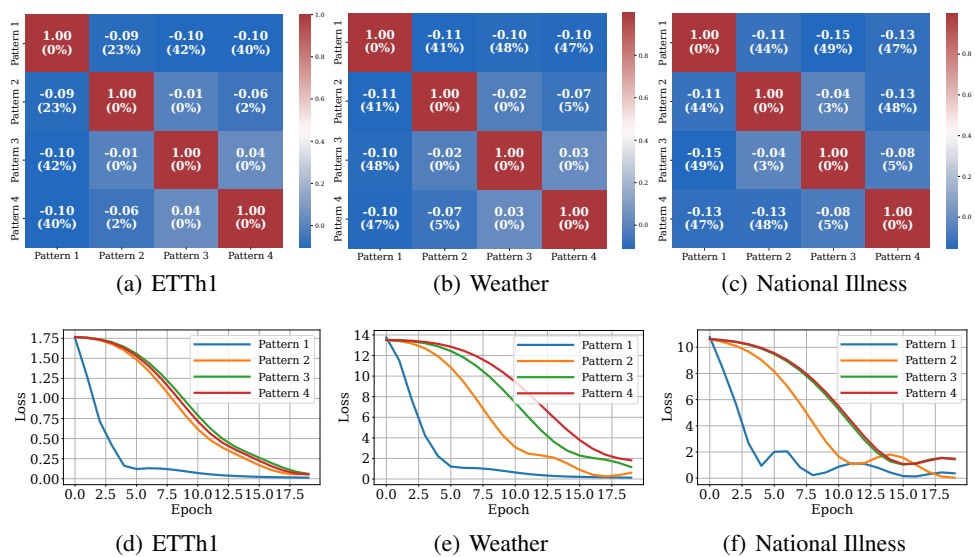

Figure 3: Gradient behaviors and early learning dynamics of decomposed patterns $\{P_{\text{in}}^{(i)}\}_{i=1}^{N}$ across real-world datasets. **(Top)** Gradients of the dominant pattern (Pattern 1) frequently conflict with others, as shown by negative cosine similarities (percentages = fraction of training steps with negative alignment), demonstrating gradient interference in single-loss training. **(Bottom)** Independent training of each component shows that dominant patterns converge much faster, while others lag behind, illustrating spectral bias toward simpler structures.

which leads to the Gram matrix

$$K_{ij} = \nabla_\theta f_\theta(\mathbf{z}_i)^\top \nabla_\theta f_\theta(\mathbf{z}_j). \tag{3}$$

Here, $f_\theta(\mathbf{z})$ represents the model's output parameterized by weights $\theta$. This Gram matrix $K$ characterizes the similarity between two inputs in terms of how they influence model updates. The eigenvalues of $K$ reveal which directions in function space are learned more quickly. Larger eigenvalues correspond to smooth, low-frequency functions, while smaller eigenvalues are associated with high-frequency or complex patterns (Valle-Perez et al., 2018; Cao et al., 2019; Arora et al., 2019). This provides a theoretical explanation for the phenomenon of spectral bias, where neural networks prioritize low-frequency structures in early training (Rahaman et al., 2019).

In the context of time-series forecasting, spectral bias becomes particularly problematic, as signals often consist of overlapping components (e.g., trend, seasonality, residual) with differing complexity (Ackaah-Gyasi et al., 2023; Das et al., 2020). A single-loss objective tends to allocate model capacity disproportionately to dominant, low-frequency patterns, leading to poor learning of higher-frequency yet predictive components. This effect can be observed empirically in Figure 3, where dominant components converge rapidly while others lag behind. From the NTK perspective, this imbalance arises because the eigenvalue spectrum is skewed: directions corresponding to smooth components dominate the training dynamics, while residual components are associated with eigen-directions of much smaller magnitude.

These insights suggest that mitigating spectral bias requires flattening the eigenvalue spectrum so that heterogeneous components are optimized at comparable rates. In the following section, we show that Separable Training achieves exactly this: by isolating structural components and assigning each a dedicated predictor, it reduces cross-term interference and balances the NTK spectrum across components, thereby providing a principled explanation for the robustness of structure-aligned learning.

## 3 PROPOSED METHOD

After introducing the limitation of the single-loss training (cf. Eq. 4), we solve the limitation by i) iteratively decomposing time-series into multiple components — we resort to existing decomposition methods — and ii) training a model for each component independently. Our theoretical analyses

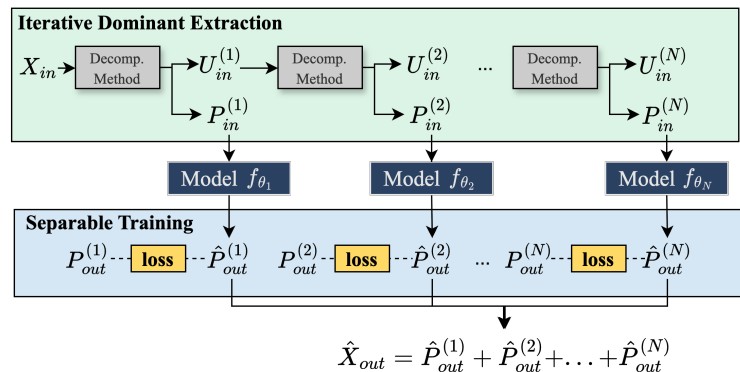

Figure 4: Overall architecture of SALT. The IDE module progressively extracts dominant and predictable components $P_{\text{in}}^{(i)}$ from the input sequence using a base decomposition method, such as STL or STR. Each component is independently forecast using a dedicated model in the Separable Training phase. The final output $\hat{X}_{\text{out}}$ is obtained by aggregating all individual predictions.

show that the proposed training method still works as intended even when the decomposition method does not completely remove noisy components.

### 3.1 MOTIVATIONS & PROBLEM DEFINITION

Most existing forecasting models adopt a single-loss paradigm, minimizing prediction error in the original time-series domain:

$$\theta \leftarrow \theta - \eta \nabla_\theta \mathcal{L}(X_{\text{out}}, \hat{X}_{\text{out}}), \tag{4}$$

where $\mathcal{L}$ is typically the mean squared error (MSE), $X_{\text{out}}$ is the ground-truth output, and $\hat{X}_{\text{out}} = f_\theta(X_{\text{in}})$ is the model prediction. While simple and widely used, this approach forces the model to jointly optimize patterns of very different frequency, scale, and semantics within a single objective. This leads to two characteristic difficulties.

First, spectral bias — the tendency of neural networks to prioritize low-frequency signals (Rahaman et al., 2019; Jacot et al., 2018) — is especially pronounced in forecasting (Das et al., 2020), where dominant trends or seasonalities are optimized much faster than residual fluctuations (see Fig. 3, Bottom). Second, gradient interference, well-studied in multi-task learning (Chai et al., 2022), also manifests in time-series models: heterogeneous structures such as smooth long-term trends and noisy residual components often yield conflicting update directions, as observed in time-series forecasting (Feng et al., 2024; Fan et al., 2021) (Fig. 3, Top). Together, these phenomena slow convergence and degrade generalization, particularly under complex or non-stationary dynamics (Assandri et al., 2023; Baidya & Lee, 2024). Our empirical observations therefore confirm that single-loss optimization underutilizes structural diversity in time-series data.

To formalize this structure, we represent the input sequence as a sum of $N$ interpretable components and a residual:

$$X_{\text{in}} = \sum_{i=1}^{N} P_{\text{in}}^{(i)} + U_{\text{in}}^{(N)}, \tag{5}$$

where each $P_{\text{in}}^{(i)}$ denotes a distinct structural pattern (not necessarily fixed as trend or seasonality), and $U_{\text{in}}^{(N)}$ collects the remaining unpredictable variation. This decomposition is not unique, but it provides a principled view of time-series as multi-component signals with heterogeneous learning dynamics. It also establishes the foundation for analyzing why optimizing each component separately is theoretically beneficial, which is the central focus of this work.

## 3.2 Overall Workflow

As illustrated in Figure 4, SALT follows a three-stage, structure-aligned pipeline designed to mitigate gradient interference and spectral bias.

1. **Iterative Dominant Extraction (IDE)** decomposes the input series into multiple dominant and predictable components (highlighted in light green box).
2. **Separable Training** independently optimizes a dedicated predictor for each component (highlighted in blue box).
3. **Aggregation** sums the predictions to form the final output, providing both improved robustness and interpretability by attributing the forecast to individual structural sources.

## 3.3 Iterative Dominant Extraction (IDE)

To implement the proposed regime, we adopt a lightweight preprocessing step that decomposes an input time series $X_{\text{in}}$ into dominant components $\{P_{\text{in}}^{(i)}\}_{i=1}^{N}$ and a residual $U_{\text{in}}^{(N)}$. We emphasize that the specific decomposition method is not central to our framework: IDE serves only as a practical way to expose structured variation that can then be trained separately. In particular, the extracted components need not be perfectly clean. Even when $P_{\text{in}}^{(i)}$ contains small residual noise, our theoretical results (see Theorem 2) show that Separable Training still yields improved forecasting performance under mild conditions commonly satisfied by methods such as STL or STR.

The process is iterative: at each step, a base decomposition method $\mathcal{D}$ (e.g., STL, STR, or simple frequency filters) is applied to the current residual to extract the most predictable structure:

$$P_{\text{in}}^{(i)} := \mathcal{D}(U_{\text{in}}^{(i-1)}), \quad U_{\text{in}}^{(i)} := U_{\text{in}}^{(i-1)} - P_{\text{in}}^{(i)}. \tag{6}$$

The iteration continues until a predefined number $N$ is reached or the residual energy becomes negligible. As shown in Figures 14 and 16, the residual becomes increasingly noise-like, supporting the intuition that IDE progressively removes predictable structure.

A detailed description of the procedure and selection criteria for different base methods is provided in Algorithm 1 (cf. Appendix D). In practice, IDE is one of many possible choices: any reasonable decomposition strategy can be plugged into our framework without affecting the theoretical conclusions.

## 3.4 Separable Training

Single-loss training jointly optimizes all temporal structures under a shared objective. While simple, this unified treatment often leads to gradient interference and inefficient use of model capacity, as dominant low-frequency patterns overshadow higher-frequency but informative structures. To avoid these issues, we adopt Separable Training, a regime in which each decomposed component is optimized with its own loss. This idea is not tied to a specific decomposition procedure (e.g., IDE); rather, any reasonable structural partition can be used.

Formally, let $X_{\text{in}}$ be decomposed into $N$ components $\{P_{\text{in}}^{(i)}\}_{i=1}^{N}$. For each $P_{\text{in}}^{(i)}$, we train a dedicated predictor $f_{\theta_i}$ by minimizing a component-wise loss:

$$\theta_i \; \leftarrow \; \theta_i - \eta \nabla_{\theta_i} L\big(P_{\text{out}}^{(i)}, \hat{P}_{\text{out}}^{(i)}\big), \tag{7}$$

where $\hat{P}_{\text{out}}^{(i)} = f_{\theta_i}(P_{\text{in}}^{(i)})$. In this setup, each component is optimized independently, cross-term gradient conflicts are eliminated, and each model can specialize to the frequency range or structure it receives. From an error decomposition view, this reduces variance and removes unstable interactions, from an NTK perspective, it balances convergence by flattening the eigenvalue spectrum.

After training, the overall forecast is simply the sum of component-wise predictions:

$$\hat{X}_{\text{out}}^{\text{SALT}} = \sum_{i=1}^{N} \hat{P}_{\text{out}}^{(i)}, \qquad \hat{X}_{\text{out}}^{\text{single}} = f(X_{\text{in}}). \tag{8}$$

This contrasts with single-loss training, where a single predictor $f$ must simultaneously fit all structures. Separable Training thus provides a principled way to align the optimization objective with the structured nature of time-series.

### 3.5 THEORETICAL JUSTIFICATIONS

To support the design of SALT, we provide two complementary theoretical justifications — one from the error reduction perspective based on structured-wise decoupling, and the other from the learning dynamics perspective using the Neural Tangent Kernel (NTK) theory.

#### 3.5.1 ERROR REDUCTION DUE TO NO GRADIENT INTERFERENCE

**Notation:** Theorem 1 builds upon the SALT architecture introduced in Figure 4. Each component $P_{in}^{(i)}$ is fed into a dedicated forecasting model $f_{\theta_i}$ to produce $\hat{P}_{out}^{(i)}$, the predicted output for that structure. The final prediction under the SALT framework is obtained by aggregating the independent predictions as in $\hat{X}_{\text{out}}^{\text{SALT}}$.

The following theorem compares the expected error between these two prediction strategies when the gradient interference problem occurs.

**Theorem 1 (Error Reduction via Structure-wise Decoupling).** *Suppose that the gradient interference problem occurs for the single-loss — we empirically justify this in Figure 3 — then their expected squared errors satisfy:*

$$\mathbb{E}\left[\left\|\hat{X}_{out}^{SALT} - X_{out}\right\|^2\right] \leq \mathbb{E}\left[\left\|\hat{X}_{out}^{single} - X_{out}\right\|^2\right]. \tag{9}$$

**Remark:** This result shows that separating training across structural components reduces total prediction error. In single-loss optimization, destructive gradient interference prevents convergence to the optimum, whereas component-wise specialization avoids such conflicts and yields more stable error reduction (see Appendix B.1 for the detailed proof).

#### 3.5.2 NO SPECTRAL BIAS DUE TO EQUAL EMPHASES ON COMPONENTS

**Notation:** Theorem 2 considers a kernel-based analysis of the inputs used in forecasting. Let $\mathbf{x}_l$ denote the original observed input at index $l$, which includes both predictable structure and noise. Here, the index $l$ corresponds to the time step in the input time-series. Let $\mathbf{p}_l$ denote the structured representation extracted from $\mathbf{x}_l$ using a decomposition method (e.g., STL or frequency-filtered), which aims to isolate the predictable patterns from the noisy input.

We define the input sequence as $X_{\text{in}} = [\mathbf{x}_1, \ldots, \mathbf{x}_n]^\top$, and the structured input sequence as $P_{\text{in}}^{(i)} = [\mathbf{p}_1^{(i)}, \ldots, \mathbf{p}_n^{(i)}]^\top$. We construct two Gram matrices:

1. $K^{(\text{single})}$ uses the raw noisy inputs $\{\mathbf{x}_l\}_{l=1}^n$;

2. $K^{(\text{SALT})}$ uses the denoised or structured inputs $\{\mathbf{p}_l^{(i)}\}_{l=1}^n$.

Both matrices are computed using a common positive semidefinite kernel $\kappa(\cdot, \cdot)$ that decreases monotonically with squared Euclidean distance (e.g., Gaussian or Laplacian kernel). The theorem compares their eigenvalue spectra to quantify how decomposition improves the spectral concentration of the input representation.

**Theorem 2 (NTK Gram Matrix Eigenvalue Comparison over Decomposition Steps).** *Let $\{\mathbf{x}_l\}_{l=1}^n \subset \mathbb{R}^d$ be observed input sequences modeled as:*

$$\mathbf{x}_l = \sum_{i=1}^N \mathbf{p}_l^{(i)} + \zeta_l, \tag{10}$$

*where each $\mathbf{p}_l^{(i)}$ denotes the $i$-th structured component extracted by a decomposition process (e.g., trend, seasonality), and $\zeta_l \sim \mathcal{N}(0, \sigma^2 I_d)$ represents residual noise. We model the residual noise as i.i.d. Gaussian for analytical convenience; the proof only relies on zero-mean, independent, isotropic noise with finite variance, so Gaussianity is a sufficient but not necessary condition. Let $\mathbf{p}_l := \sum_{i=1}^N \mathbf{p}_l^{(i)}$ be the aggregated structured representation. Define the following NTK Gram*

*matrices using a positive semidefinite kernel $\kappa(\cdot, \cdot)$ that is monotonically decreasing in squared Euclidean distance:*

$$K^{(\text{SALT})} := [\kappa(\mathbf{p}_l, \mathbf{p}_m)]_{l,m=1}^{n}, \tag{11}$$

$$K^{(\text{single})} := [\kappa(\mathbf{x}_l, \mathbf{x}_m)]_{l,m=1}^{n}. \tag{12}$$

*Then, under mild assumptions on the decomposition quality and noise independence (see Appendix B.2), The eigenvalues of these matrices satisfy:*

$$\lambda_s^{(\text{SALT})} > \lambda_s^{(\text{single})}, \quad \text{for all } s = 1, \ldots, n. \tag{13}$$

**Remark:** This theorem indicates that decomposition improves the spectral concentration of NTK representations. By isolating structured patterns, SALT produces Gram matrices with larger eigenvalues, which correspond to faster and balanced learning across components. This mitigates spectral bias, ensuring that high-frequency or less dominant structures are not neglected during training. As decomposition depth increases, our corollaries (Appendix B.3) show that eigenvalues accumulate, capturing richer structural information and leading to lower prediction error (cf. Figure 5).

**Summary of theoretical justifications:** We provide two complementary theoretical justifications for this approach. Theorem 1 establishes that Separable Training achieves a provable upper bound on the total prediction error by eliminating destructive gradient interference. Theorem 2 shows that, even in the simplest case ($i = 1$), SALT yields NTK Gram matrices with strictly larger eigenvalues than those from raw inputs, reflecting faster and more balanced convergence across components. Moreover, as the number of decomposition steps increases, this spectral advantage accumulates, allowing the model to capture richer structural information and further reduce prediction error. Taken together, these results provide a principled explanation of why structure-aligned learning improves both stability and generalization compared to single-loss optimization.

## 4 EXPERIMENTS

**Experimental settings:** We list all the descriptions of datasets, baselines and detailed experimental settings in Appendix G.4. For implementation details, STL and STR decomposition are applied to the training and validation series to obtain cleaner structural components for model learning. At test time, decomposition is conducted in a causal rolling manner, where each forecast step relies only on its preceding observations for component extraction. This procedure naturally reflects a realistic forecasting scenario and ensures consistent evaluation.

### 4.1 MAIN RESULTS

We evaluate SALT on 9 real-world time-series datasets across 6 state-of-the-art forecasting models and 4 prediction horizons (with ILL using 4 shorter horizons). Table 1 reports the average performance across all horizons. We observe that SALT consistently improves the base model's forecasting accuracy in the majority of settings. On average, SALT achieves over 10% MSE reduction across models, with particularly strong gains on datasets with high non-stationarity such as ETTh1 and ILL (e.g., up to 26.3% and 29.2% MSE improvement, respectively).

Notably, even strong baselines like N-BEATS, N-HiTS, and PatchTST benefit from SALT, indicating that the gains are not due to weak base models but rather due to SALT' ability to better align with the structured nature of time-series data. These empirical results validate our hypothesis that component-wise learning mitigates gradient interference and spectral bias — core challenges in time-series forecasting. Detailed results for each horizon are provided in the Appendix G.2.

### 4.2 ABLATION STUDY AND SENSITIVITY ANALYSIS

Additional ablation studies are provided in Appendix G.1. Table 11 evaluates different decomposition strategies within SALT and shows consistent improvements across all architectures and datasets, confirming robustness to the choice of base method (STR, STL, frequency-based). Table 12 examines the effect of input noise for fair comparison; even when single-loss models are trained on

Table 1: Experimental results on 9 benchmark datasets. Results where the SALT method improved the performance of a base model are highlighted in **bold**, and the best performance w.r.t. a pair of dataset among all is highlighted in *italic*. The entire results is in the Appendix G.2.

| Datasets | Weather | | Exchange | | ETTh1 | | ETTm1 | | ETTh2 | | ETTm2 | | Electricity | | Traffic | | ILL | |
|---|---|---|---|---|---|---|---|---|---|---|---|---|---|---|---|---|---|---|
| | MSE | MAE | MSE | MAE | MSE | MAE | MSE | MAE | MSE | MAE | MSE | MAE | MSE | MAE | MSE | MAE | MSE | MAE |
| N-BEATS | 0.262 | 0.281 | 0.481 | 0.455 | 0.432 | 0.441 | 0.383 | 0.398 | 0.363 | 0.409 | 0.294 | 0.345 | 0.197 | 0.300 | 0.461 | 0.321 | 2.269 | 1.012 |
| SALT | **0.246** | **0.278** | **0.444** | **0.443** | **0.385** | **0.423** | **0.317** | **0.357** | **0.353** | **0.394** | **0.282** | **0.330** | **0.194** | **0.296** | **0.454** | **0.315** | **1.958** | **0.964** |
| Imp.(Avg.) | 6.27% | 1.51% | 7.72% | 2.57% | 10.7% | 3.80% | 17.2% | 10.3% | 2.75% | 3.66% | 3.83% | 4.15% | 1.58% | 1.07% | 1.54% | 1.86% | 13.6% | 4.75% |
| N-HiTS | 0.248 | 0.273 | 0.349 | 0.380 | 0.411 | 0.426 | 0.368 | 0.383 | 0.337 | 0.393 | 0.279 | 0.329 | 0.185 | 0.287 | 0.452 | 0.311 | 2.051 | 0.925 |
| SALT | **0.235** | **0.259** | **0.316** | *0.363* | **0.396** | **0.422** | **0.350** | **0.375** | **0.334** | **0.380** | **0.269** | **0.314** | **0.173** | **0.277** | **0.428** | **0.303** | **1.917** | **0.888** |
| Imp.(Avg.) | 5.12% | 5.32% | 9.43% | 4.44% | 3.50% | 0.87% | 4.66% | 1.98% | 0.82% | 3.01% | 3.47% | 4.83% | 6.59% | 3.46% | 5.22% | 2.37% | 6.51% | 3.97% |
| DLinear | 0.246 | 0.300 | 0.296 | 0.377 | 0.422 | 0.437 | 0.357 | 0.378 | 0.431 | 0.446 | 0.267 | 0.331 | 0.166 | 0.263 | 0.433 | 0.295 | 2.169 | 1.041 |
| SALT | **0.232** | **0.290** | *0.290* | **0.366** | **0.329** | **0.397** | **0.315** | **0.358** | **0.413** | **0.432** | **0.257** | **0.324** | **0.156** | **0.260** | *0.329* | **0.286** | **1.619** | **0.869** |
| Imp.(Avg.) | 5.67% | 3.14% | 2.08% | 2.89% | 22.1% | 8.99% | 11.6% | 5.28% | 4.19% | 3.16% | 3.54% | 2.02% | 6.04% | 1.15% | 24.0% | 2.99% | 25.3% | 16.4% |
| PatchTST | 0.225 | 0.263 | 0.463 | 0.451 | 0.413 | 0.434 | 0.351 | 0.380 | 0.330 | 0.380 | 0.256 | 0.315 | 0.159 | 0.252 | 0.391 | 0.264 | 1.223 | 0.806 |
| SALT | *0.212* | *0.253* | **0.360** | **0.410** | *0.316* | *0.383* | *0.301* | *0.353* | *0.327* | *0.374* | *0.255* | *0.313* | *0.153* | *0.248* | **0.389** | *0.259* | *1.174* | *0.747* |
| Imp.(Avg.) | 5.96% | 4.01% | 22.2% | 9.02% | 23.3% | 11.5% | 14.2% | 7.14% | 1.04% | 1.44% | 0.11% | 0.71% | 3.60% | 1.51% | 0.38% | 1.52% | 4.00% | 7.38% |
| iTransformer | 0.257 | 0.277 | 0.360 | 0.403 | 0.454 | 0.447 | 0.407 | 0.409 | 0.383 | 0.406 | 0.288 | 0.332 | 0.178 | 0.269 | 0.428 | 0.282 | 2.085 | 0.987 |
| SALT | **0.229** | **0.264** | **0.325** | **0.384** | **0.351** | **0.410** | **0.335** | **0.386** | **0.374** | **0.397** | **0.285** | **0.328** | **0.169** | **0.262** | **0.392** | **0.280** | **1.633** | **0.889** |
| Imp.(Avg.) | 11.6% | 6.03% | 10.5% | 5.39% | 24.1% | 9.44% | 18.2% | 6.16% | 1.50% | 0.18% | 0.17% | 0.22% | 4.62% | 0.37% | 7.65% | 0.44% | 21.9% | 11.1% |
| TimeMixer | 0.240 | 0.271 | 0.420 | 0.426 | 0.446 | 0.440 | 0.381 | 0.395 | 0.364 | 0.395 | 0.275 | 0.323 | 0.182 | 0.272 | 0.484 | 0.297 | 1.799 | 0.892 |
| SALT | **0.233** | **0.264** | **0.388** | **0.414** | **0.328** | **0.392** | **0.308** | **0.364** | **0.353** | **0.390** | **0.267** | **0.316** | **0.162** | **0.259** | **0.407** | **0.292** | **1.273** | **0.770** |
| Imp.(Avg.) | 2.97% | 2.66% | 7.63% | 2.84% | 26.3% | 10.8% | 19.1% | 7.99% | 2.97% | 1.22% | 2.75% | 1.97% | 11.0% | 4.72% | 15.9% | 1.78% | 29.2% | 13.6% |

denoised inputs, they fail to match SALT, demonstrating that its gains arise from structural learning rather than input quality. We also compare against ensemble-like baselines, and SALT achieves superior accuracy despite using the same underlying backbones, validating the benefit of its Separable Training design.

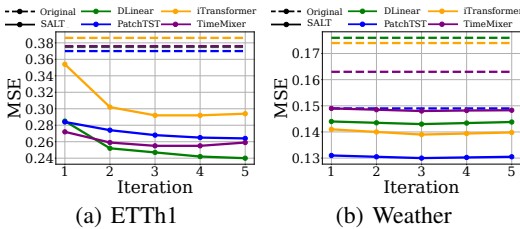

(a) ETTh1 (b) Weather

Figure 5: Sensitivity to iteration number $N$ of Separable Training. More results in Appendix G.1.

**Sensitivity on IDE iterations:** Figure 5 demonstrates how MSE decreases progressively as the number of iterations in the iterative residual decomposition and Separable Training increases within the SALT framework. Datasets like ETTh1 repeated iterations consistently reduce MSE, improving forecasting accuracy compared to the original models (dotted line), highlighting the effectiveness of SALT. The framework iteratively learns from predictable patterns extracted at each decomposition stage, as shown in Figure 3 (Bottom) capturing the data's underlying structures. This process refines the model's ability to separate patterns from noise, addressing the bias-variance tradeoff more efficiently.

## 5 CONCLUSION

In this work, we present SALT for time-series forecasting that addresses two core optimization challenges: gradient interference and spectral bias. By decomposing time-series data into interpretable components and training each with a dedicated predictor, SALT enables more effective learning of complex temporal structures. Our theoretical analyses, based on structure-wise decoupling perspective and NTK spectral insights, provide strong justifications for the benefits of this separable approach. Extensive experiments on nine benchmark datasets and six state-of-the-art forecasting models demonstrate that SALT consistently improves both predictive accuracy and training stability. Given its modular design and empirical robustness, SALT offers a principled and extensible solution for modern time-series forecasting models.

**Ethics statement**  This work focuses exclusively on methodological and theoretical contributions to time-series forecasting. All experiments were conducted on publicly available benchmark datasets that contain only numerical time-series data without personal or sensitive information. No human subjects, personally identifiable data, or sensitive attributes were involved. We do not foresee any direct ethical concerns or potential societal harms arising from this research.

**Reproducibility Statement**  To ensure the reproducibility and completeness of this paper, we make our code available at `https://drive.google.com/drive/folders/1dN_a1ZReKH0AZe4R_RxbP5Sbtj5HNydB?usp=sharing`. We give details on our experimental protocol in the Appendix G.7.

**The Use of Large Language Models (LLMs)**  We used ChatGPT as a writing assistant for polishing language and checking notation. No research ideas or content generation were conducted by LLMs.

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

# A   APPENDIX CONTENTS

# B   PROOFS

## B.1   PROOF OF THEOREM 1

**Theorem 1 (Error Reduction via Structure-wise Decoupling).** *Let $X = \sum_{i=1}^{N} P^{(i)}$ be a structured target composed of $K$ interpretable components. Let us adopt the following notations:*

- *$\hat{X}_{out}^{SALT} := \sum_{i=1}^{N} \hat{P}_{out}^{(i)}$, where each $\hat{P}_{out}^{(i)}$ is trained independently on $P_{in}^{(i)}$ to predict $P_{out}^{(i)}$;*

- *$\hat{X}_{out}^{single} := f(X_{in})$,the output of a conventional single-loss model trained to predict the full target $X_{out}$.*

*We also define the component-wise error as follows:*

$$\epsilon^{(i)} := \hat{P}_{out}^{(i)} - P_{out}^{(i)}, \quad \acute{\epsilon}^{(i)} := \acute{P}_{out}^{(i)} - P_{out}^{(i)}, \tag{14}$$

*where $\acute{P}_{out}^{(i)}$ is the orthogonal projection of $\hat{X}_{out}^{single}$ onto the subspace spanned by $P_{out}^{(i)}$. The projection is given by:*

$$\acute{P}_{out}^{(i)} := \frac{\langle \hat{X}_{out}^{single}, P_{out}^{(i)} \rangle}{\langle P_{out}^{(i)}, P_{out}^{(i)} \rangle} \cdot P_{out}^{(i)}, \tag{15}$$

which corresponds to the least-squares approximations of $\hat{X}_{out}^{single}$ along the direction of $P_{out}^{(i)}$. This projection follows the standard definition of the least-squares projection onto a one-dimensional subspace, as commonly introduced in classical linear algebra textbooks (Strang, 2000).

Under these assumptions, the following inequality holds:

$$\mathbb{E}\left[\left\|\hat{X}_{out}^{SALT} - X_{out}\right\|^2\right] \leq \mathbb{E}\left[\left\|\hat{X}_{out}^{single} - X_{out}\right\|^2\right]. \tag{16}$$

*Proof.* From the linearity of $X_{\text{out}} = \sum_i^N P_{\text{out}}^{(i)}$, the prediction errors are

$$\hat{X}_{\text{out}}^{\text{SALT}} - X_{\text{out}} = \sum_i^N \epsilon^{(i)}, \quad \hat{X}_{\text{out}}^{\text{single}} - X_{\text{out}} = \sum_i^N \acute{\epsilon}^{(i)}. \tag{17}$$

By expansion,

$$\mathbb{E}\left[\left\|\hat{X}_{\text{out}}^{\text{SALT}} - X_{\text{out}}\right\|^2\right] = \sum_i \mathbb{E}[\|\epsilon^{(i)}\|^2], \tag{18}$$

$$\mathbb{E}\left[\left\|\hat{X}_{\text{out}}^{\text{single}} - X_{\text{out}}\right\|^2\right] = \sum_i \mathbb{E}[\|\acute{\epsilon}^{(i)}\|^2] + \sum_{j \neq k} \mathbb{E}[\langle \acute{\epsilon}^{(j)}, \acute{\epsilon}^{(k)} \rangle]. \tag{19}$$

Since each structure-specific error $\epsilon^{(i)}$ arises from a completely disjoint predictor with no shared inputs or parameters, the cross-terms $\mathbb{E}[\langle \epsilon^{(j)}, \epsilon^{(k)} \rangle]$ vanish for $j \neq k$.

As a result, the total squared error decomposes cleanly as follows:

$$\mathbb{E}\left[\left\|\hat{X}_{\text{out}}^{\text{SALT}} - X_{\text{out}}\right\|^2\right] = \sum_i \mathbb{E}[\|\epsilon^{(i)}\|^2]. \tag{20}$$

By the following two reasonable assumptions,

1. Gradient interference: $\mathbb{E}[\langle \acute{\epsilon}^{(j)}, \acute{\epsilon}^{(k)} \rangle] > 0$,

2. Component-wise efficiency: $\mathbb{E}[\|\epsilon^{(i)}\|^2] \leq \mathbb{E}[\|\acute{\epsilon}^{(i)}\|^2]$,

we conclude that

$$\mathbb{E}\left[\left\|\hat{X}_{\text{out}}^{\text{SALT}} - X_{\text{out}}\right\|^2\right] \leq \mathbb{E}\left[\left\|\hat{X}_{\text{out}}^{\text{single}} - X_{\text{out}}\right\|^2\right]. \tag{21}$$

$\square$

**Remark.** This result formalizes the theoretical benefit of structure-wise decoupling in time-series forecasting. In SALT, each component predictor $\hat{P}^{(i)}$ is trained independently using disjoint inputs, separate model parameters, and distinct loss objectives. This architectural independence ensures that cross-term interference — often present in single-loss optimization — is structurally eliminated, allowing the total MSE error to decompose into a sum of separable errors. This yields a provably tighter error bound compared to single-loss training, especially when components differ in complexity or predictability.

**Structural Properties Used in Theorem 1.** Our theoretical result is based on the structural decoupling of separate predictors and uses a projection-based decomposition to analyze and compare conventional single-loss models and SALT in the common output space.

- **Validity of Projection.** To compare SALT with a conventional single-loss model, we project the output $\hat{X}_{\text{out}}^{\text{single}}$ onto the same set of structural components $\{P_{\text{out}}^{(i)}\}$ using least-squares decomposition. This projection is mathematically valid regardless of whether the original components are orthogonal or not, since we treat each subspace independently and apply standard linear regression to extract $\tilde{P}_{\text{out}}^{(i)}$. The orthogonality assumption is not necessary when assessing prediction errors under Separable Training (e.g., SALT).

- **Structural Elimination of Cross-Terms in SALT.** In SALT, each predictor $\hat{P}_{out}^{(i)}$ is trained using fully disjoint inputs, separate parameter sets, and independent optimization. Therefore, the resulting errors $\epsilon^{(i)}$ are orthogonal in expectation by design:

$$\mathbb{E}[\langle \epsilon^{(i)}, \epsilon^{(j)} \rangle] = 0 \quad \text{for } i \neq j. \tag{22}$$

  As a result, the total prediction error becomes a simple sum of per-component terms, eliminating the need for any statistical assumptions or independence constraints.

**Justification of Key Assumptions for Theorem 1**  To establish the inequality in Theorem 1, we rely on two key assumptions that relate to the optimization behavior of conventional model (single-loss training model) and SALT (Separable Training model). The following subsections provide theoretical motivations and empirical support for each assumption.

- **Assumption 1: Positive Cross-Terms in Conventional Models via Gradient Interference.** In contrast, conventional models trained with a single-loss and shared parameters naturally entangle component updates, often leading to correlated prediction errors. This is a direct consequence of destructive gradient interference, where optimizing one component interferes with others — especially when dominant and minor patterns compete. Such entanglement typically results in $\mathbb{E}[\langle \tilde{\epsilon}^{(i)}, \tilde{\epsilon}^{(j)} \rangle] > 0$, as supported by our gradient alignment and error correlation visualizations (Figure 3(Top)), and consistent with prior findings in multi-task learning (Liu et al., 2021; Javaloy & Valera, 2021).

- **Assumption 2: Efficiency of Separable Training.** We assume that each component-specific predictor achieves lower (or at least not worse) prediction error than the corresponding projection from the jointly trained model. This assumption is supported by both empirical results and the architectural structure of SALT. Empirically, as shown in Table 1, per-component models consistently yield lower MSE than the projected outputs from $\hat{X}_{out}^{single}$, particularly when components vary in scale, complexity, or temporal frequency. Structurally, this efficiency stems from the fact that SALT avoids gradient interference by training each component independently, using separate objectives and disjoint parameter sets. This separation enables each predictor to specialize in its respective component without being affected by conflicting optimization signals from other structures, which is a known limitation of single-loss training. While this assumption may not universally hold in all scenarios, it is both theoretically motivated and practically validated in our setting—especially under heterogeneous component conditions where decoupled learning offers significant benefits.

## B.2    PROOF OF THEOREM 2

**Theorem 2** (NTK Gram Matrix Eigenvalue Comparison over Decomposition Steps). *While Theorem 2 is stated without qualification due to the page limit in Section named Theoretical Justifications, this appendix provides a complete proof under minimal conditions. In particular, the eigenvalue ordering*

$$\lambda_s^{(SALT)} > \lambda_s^{(single)} \tag{23}$$

*holds strictly when the structured inputs $\mathbf{p}_l$ are non-trivial and the noise $\zeta_l$ is non-zero. These assumptions are mild and easily satisfied in practice, as discussed below.*

*Let $\{x_l\}_{l=1}^n \subset \mathbb{R}^d$ be observed time-series samples, where each sample is modeled as follows:*

$$\mathbf{x}_l = \sum_{i=1}^{N} \mathbf{p}_l^{(i)} + \zeta_l, \tag{24}$$

*where $\mathbf{p}_l^{(i)}$ denotes the $i$-th structured component extracted via iterative decomposition (e.g., trend, seasonality, residual patterns), and $\zeta_l \sim \mathcal{N}(0, \sigma^2 I_d)$ denotes the remaining residual noise.*

*Assume that each component satisfies*

$$\mathbf{p}_l^{(i)} = \mathbf{r}_l^{(i)} + \eta_l^{(i)}, \quad \eta_l^{(i)} \sim \mathcal{N}(0, \sigma_i^2 I_d), \tag{25}$$

where $\mathbf{r}_l^{(i)}$ is the clean structured signal and $\eta_l^{(i)}$ is its internal Gaussian perturbation, independent across $i$.

Let $K^{(i)} := [\kappa(\mathbf{p}_l^{(i)}, \mathbf{p}_m^{(i)})]$ be the NTK Gram matrix computed for component $i$, and define the SALT kernel as

$$K^{(\text{SALT})} := \sum_{i=1}^{N} K^{(i)}, \qquad K^{(\text{single})} := [\kappa(\mathbf{x}_l, \mathbf{x}_m)], \tag{26}$$

where $\kappa$ is a positive semi-definite kernel monotonically decreasing in squared Euclidean distance.

For all $s$, the following holds

$$\lambda_s^{(\text{SALT})} \geq \lambda_s^{(\text{single})}, \tag{27}$$

with strict inequality holding under mild separation between structured components and noise.

*Proof.* We begin by analyzing the expected pairwise squared distance in the single-loss setting as follows:

$$\begin{aligned}
\mathbf{x}_l = \sum_{i=1}^{N} \mathbf{p}_l^{(i)} + \zeta_l = \mathbf{p}_l + \zeta_l, &\quad \mathbf{p}_l = \sum_{i=1}^{N} \mathbf{p}_l^{(i)}, \\
\Rightarrow \mathbb{E}\left[\|\mathbf{x}_l - \mathbf{x}_m\|^2\right] &= \|\mathbf{p}_l - \mathbf{p}_m\|^2 \\
&+ \mathbb{E}[\|\zeta_l - \zeta_m\|^2] + 2\mathbb{E}[\langle \mathbf{p}_l - \mathbf{p}_m, \zeta_l - \zeta_m \rangle] \\
&= \|\mathbf{p}_l - \mathbf{p}_m\|^2 + 2\sigma^2 d,
\end{aligned} \tag{28}$$

where cross terms vanish due to independence and zero mean of $\zeta_l$.

Since $\kappa$ decreases with squared distance,

$$\mathbb{E}[\kappa(\mathbf{x}_l, \mathbf{x}_m)] < \kappa(\mathbf{p}_l, \mathbf{p}_m), \tag{29}$$

which implies

$$\mathbb{E}[K^{(\text{single})}] \preceq K^{(1)} + K^{(2)} + \cdots + K^{(N)} = K^{(\text{SALT})}. \tag{30}$$

Finally, from the theory of PSD matrix ordering, if $A \preceq B$ for PSD matrices,

$$\lambda_s(A) \leq \lambda_s(B), \quad \text{for all } s, \tag{31}$$

which proves the theorem as follows:

$$\lambda_s^{(\text{single})} \leq \lambda_s^{(\text{SALT})}. \tag{32}$$

Its strict inequality holds if the noise magnitude $\sigma^2$ is nonzero and at least one component $\mathbf{p}^{(i)}$ is nontrivial (i.e., $\mathbf{p}_l^{(i)} \neq \mathbf{p}_m^{(i)}$). $\qquad\square$

**Remark.** Theorem 2 shows that, under mild decomposition and noise assumptions, the kernel matrix constructed from structured inputs has spectrally stronger characteristics than that constructed from noisy raw inputs. Specifically, the leading eigenvalues of $K^{(\text{SALT})}$ are strictly greater than those of $K^{(\text{single})}$, due to increased pairwise similarity and reduced noise-induced distortion.

This result provides a theoretical explanation for the empirical spectral sharpening observed in SALT. By removing high-variance residuals before kernel computation, the method concentrates variance into fewer principal directions, yielding more informative representations and improving optimization dynamics in kernel-based models.

**Noise-Based Modeling and Its Spectral Implication in Theorem 2** Although the extracted representation $\mathbf{p}_j$ is not strictly noise-free, since it includes the uncertainty term $\eta_l$ — it is a more structured and predictable signal than the raw input $\mathbf{x}_l$. In our theoretical framework, we model the full observation as

$$\mathbf{x}_l = \mathbf{p}_l + \eta_l, \quad \epsilon_l = \eta_l + \zeta_l. \tag{33}$$

Herein,

- $\eta_l \sim \mathcal{N}(0, \sigma^2 I_d)$ accounts for imperfections in the decomposition (i.e., uncertainty in approximating the true signal),
- $\zeta_l \sim \mathcal{N}(0, \sigma^2 I_d)$ models residual observation noise (e.g., sensor error, environmental randomness, market fluctuations),
- hence, $\epsilon_l \sim \mathcal{N}(0, 2\sigma^2 I_d)$ is the total additive noise corrupting $\mathbf{r}_l$.

Importantly, we do not assume that $\mathbf{p}_l$ is strictly closer to the ground truth $\mathbf{r}_l$ in every realization. Rather, the assumption is made at the level of *expected squared distance*. Due to the independence and isotropic nature of $\zeta_j$, we have

$$\mathbb{E}\left[\|\mathbf{x}_l - \mathbf{x}_m\|^2\right] = \|\mathbf{p}_l - \mathbf{p}_m\|^2 + 2\sigma^2 d > \|\mathbf{p}_l - \mathbf{p}_m\|^2, \tag{34}$$

for all $l \neq m$. Since most kernels used in NTK theory (e.g., Gaussian/RBF) are monotonically decreasing in squared distance, this increase in expected distance directly implies

$$\mathbb{E}[\kappa(\mathbf{x}_l, \mathbf{x}_m)] < \kappa(\mathbf{p}_l, \mathbf{p}_m), \tag{35}$$

leading to uniformly smaller kernel matrix entries for the joint model.

This reduction in similarity leads to lower eigenvalues of $K^{(\text{single})}$ compared to $K^{(\text{SALT})}$, which we formalize in Theorem 2.

**Justification of Key Assumptions for Theorem 2.**

- **i.i.d. Gaussian Residual Noise.** We assume $\zeta_l \sim \mathcal{N}(0, \sigma^2 I_d)$ to model the unpredictable residual component after decomposition. This assumption is not arbitrary: empirical results support that the residuals extracted by our Iterative Dominant Extraction (IDE) algorithm become increasingly decorrelated over iterations. As shown in Figure 14, the residuals exhibit near-white noise behavior, evidenced by the Durbin–Watson statistic approaching 2.0 and the decay of the average autocorrelation function (ACF). This confirms that the temporal structure has been largely removed, justifying the modeling of $\zeta_l$ as i.i.d. Gaussian noise.

- **Structured Decomposition.** Our decomposition procedure separates $\mathbf{x}_l$ into a structured part $\mathbf{p}_l$ and a stochastic part $\zeta_l$. While $\mathbf{p}_l$ is not an oracle ground-truth signal, it captures interpretable and predictable structures (e.g., trend, seasonality). This is consistent with prior work in time-series analysis and representation learning (Belkin et al., 2019), where denoised or structured projections yield representations with higher spectral concentration in kernel matrices. The reduction in expected squared distance between $\mathbf{p}_l$ pairs is central to the eigenvalue comparison theorem.

- **Monotonic Kernel Property.** We assume that the kernel $\kappa$ used to compute Gram matrices is positive semidefinite and monotonically decreasing with respect to squared Euclidean distance. This property holds for commonly used kernels such as

  - *Gaussian/RBF kernel:* $\kappa(\mathbf{x}, \mathbf{y}) = \exp(-\|\mathbf{x} - \mathbf{y}\|^2/2\sigma^2)$,
  - *Laplacian kernel:* $\kappa(\mathbf{x}, \mathbf{y}) = \exp(-\|\mathbf{x} - \mathbf{y}\|_1/\sigma)$.

  These kernels are frequently used in the neural tangent kernel literature (Jacot et al., 2018) and their monotonicity is critical in showing that added noise reduces kernel similarity and shrinks the spectrum.

- **Expectation Asymmetry.** We apply the expectation operator only to the kernel values involving $\mathbf{x}_l$ (i.e., $\mathbb{E}[\kappa(\mathbf{x}_l, \mathbf{x}_m)]$), while keeping $\kappa(\mathbf{p}_l, \mathbf{p}_m)$ as a deterministic quantity. This is justified because $\mathbf{x}_l$ contains stochastic noise $\zeta_l$, whereas $\mathbf{p}_l$—though potentially noisy

due to $\eta_l$—is treated as a fixed structured representation once obtained. This leads to a **conservative comparison**: we compare the expected similarity of noisy samples to the similarity of a single realization of the structured signal (Jacot et al., 2018). Since $\zeta_l$ is zero-mean and independent of $\mathbf{p}_l$, this comparison still supports the following inequality:

$$\mathbb{E}[\kappa(\mathbf{x}_l, \mathbf{x}_m)] < \kappa(\mathbf{p}_l, \mathbf{p}_m), \tag{36}$$

making our bound stronger and analytically favorable.

### B.3 Corollaries of Theorem 2

**Corollary 1** (**Lower Error in Separable Training**). *Suppose the training errors in each eigen-direction decay exponentially at rates given by the respective NTK eigenvalues:*

$$E^{\mathrm{SALT}}(t) = \sum_{s=1}^{S} C_s \, \exp\!\big(-\lambda_s^{\mathrm{SALT}} \, t\big), \tag{37}$$

$$E^{\mathrm{single}}(t) = \sum_{s=1}^{S} C_s \, \exp\!\big(-\lambda_s^{\mathrm{single}} \, t\big), \tag{38}$$

*where $C_s$ reflects the initial error magnitude or alignment in the $s$-th eigen-direction. If $\lambda_s^{\mathrm{SALT}} > \lambda_s^{\mathrm{single}}$ for all $s$ (cf. Theorem 1), then for all $t > 0$,*

$$E^{\mathrm{SALT}}(t) \ \leq \ E^{\mathrm{single}}(t). \tag{39}$$

**Corollary 2** (**Faster and More Comprehensive Learning**). *Let us define the total learned information at time $t$ in terms of how much of the initial error has been reduced:*

$$I^{\mathrm{SALT}}(t) = \sum_{s=1}^{S} \big(1 - \exp(-\lambda_s^{\mathrm{SALT}} \, t)\big), \tag{40}$$

$$I^{\mathrm{single}}(t) = \sum_{s=1}^{S} \big(1 - \exp(-\lambda_s^{\mathrm{single}} \, t)\big). \tag{41}$$

*Under the same assumption that $\lambda_s^{\mathrm{SALT}} > \lambda_s^{\mathrm{single}}$, we have for all $t > 0$:*

$$I^{\mathrm{SALT}}(t) \ > \ I^{\mathrm{single}}(t). \tag{42}$$

**Corollary 3** (**Lower Error in Multivariate Separable Training**). *For multivariate (or multi-dimensional) time-series data, the above properties hold under essentially the same conditions:*

$$E^{\mathrm{SALT}}(t) \ \leq \ E^{\mathrm{single}}(t), \quad \forall \, t > 0. \tag{43}$$

## C Theoretical Analysis

### C.1 NTK-based analysis of Single vs. SALT

To analyze the learning dynamics of different training strategies, we leverage the NTK framework, which captures the evolution of gradient similarities during training. Given a model output $f_\theta(x)$ parameterized by $\theta$, the NTK Gram matrix is defined as:

$$K_{ij} = \nabla_\theta f_\theta(\mathbf{z}_i)^\top \nabla_\theta f_\theta(\mathbf{z}_j), \tag{44}$$

where $\mathbf{z}_i, \mathbf{z}_j$ are input data points and $\nabla_\theta f_\theta(\mathbf{z})$ denotes the gradient of the model output with respect to its parameters. The NTK matrix quantifies how parameter updates propagate across different samples, revealing the extent to which different regions of the input space influence each other during training.

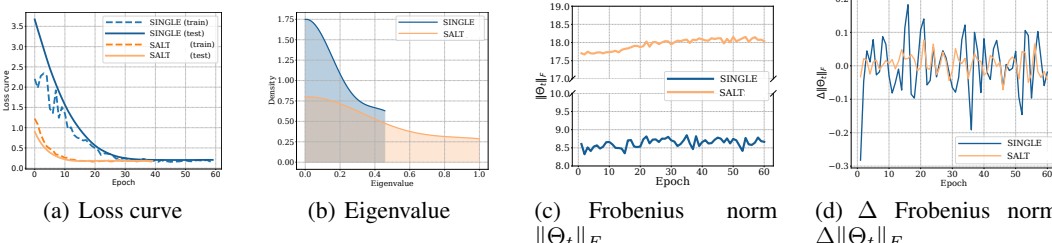

(a) Loss curve     (b) Eigenvalue     (c) Frobenius norm $\|\Theta_t\|_F$     (d) $\Delta$ Frobenius norm $\Delta\|\Theta_t\|_F$

Figure 6: Comparison of SINGLE and SALT training. (a) SALT achieves lower loss and faster convergence. (b) Its eigenvalues are more uniformly distributed, improving representation learning. (c) SALT maintains higher and more stable NTK Frobenius norms. (d) SINGLE shows abrupt changes in optimization, while SALT ensures smoother training dynamics.

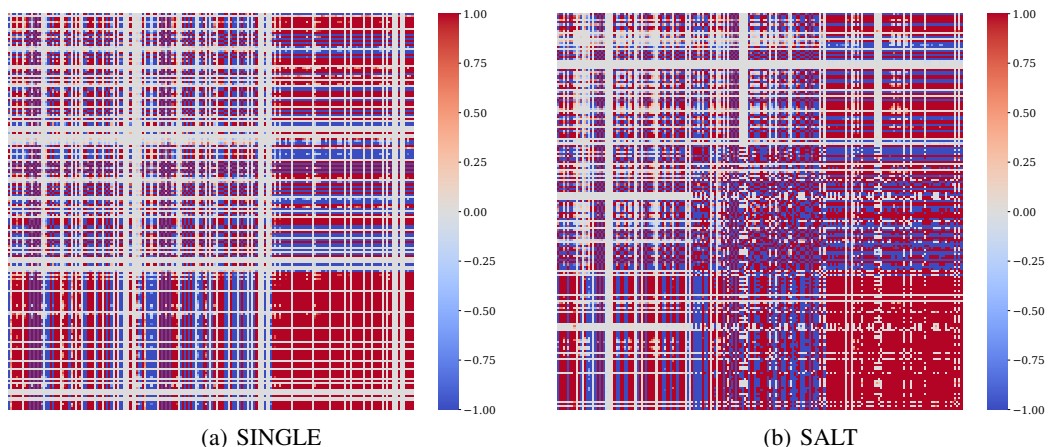

(a) SINGLE           (b) SALT

Figure 7: NTK (Neural Tangent Kernel) Gram matrix visualization. The $x$-axis and $y$-axis represent different input samples $\mathbf{z}_i, \mathbf{z}_j$, while the color intensity corresponds to the similarity between their gradient updates, defined as $K_{ij} = \nabla_\theta f_\theta(\mathbf{z}_i)^\top \nabla_\theta f_\theta(\mathbf{z}_j)$. More visualizations, including NTK Gram matrices on real-world datasets, are provided in Appendix I.3

To ensure a fair comparison between single training (SINGLE) and SALT under identical training conditions, we conducted synthetic experiments designed to reflect real-world time-series complexities. The dataset includes multiple components, such as trend, seasonality, and added Gaussian noise, along with other noise types like constant noise. This allowed us to evaluate the robustness of both training methods in handling diverse temporal structures. The experimental setup, including dataset generation details and hyperparameter configurations, is provided in Appendix G.8 for further reference.

Figure 7 visualizes the NTK Gram matrices for the Single Training and SALT models. The x-axis and y-axis represent different input samples, and the color intensity corresponds to the magnitude of the kernel entries $K_{ij}$, indicating the strength of gradient interactions between them.

In the Single Training case (Figure 7 a), large near-zero regions (gray areas) indicate that certain data points do not effectively share gradient information with others, leading to suboptimal learning of complex temporal patterns. In contrast, the NTK structure in SALT (Figure 7 b) is significantly denser, with fewer near-zero gradient regions. This suggests that SALT facilitates better gradient propagation, ensuring that both low-frequency and high-frequency components are effectively captured.

Overall, these results demonstrate that SALT enhances representational diversity and learning efficiency by improving gradient information flow across different time scales. This aligns with previ-

---

**Algorithm 1:** Iterative Dominant Extraction method

---

**Input:** Input time-series data $X_{\text{in}}$, residual iteration number $max\_iter = N$

1 $i \leftarrow 1$;
2 Decompose $X_{\text{in}}$ into $P_{\text{in}}^{(1)}$ and $U_{\text{in}}^{(1)}$;
3 $i \leftarrow 2$;
4 **while** $i < max\_iter$ **do**
5     Decompose $U_{\text{in}}^{(i-1)}$ into $P_{\text{in}}^{(i)}$ and $U_{\text{in}}^{(i)}$;
      /* Using one of (STR, STL, Freq.)                               */
6     $i \leftarrow i + 1$;
7 **return** $\{P_{\text{in}}^{(i)}\}_{i=1}^{N}$

---

ous findings in decomposition-based forecasting models Zhou et al. (2022), while further extending their benefits through separable optimization.

In Figure 6, we compare single training (SINGLE) and Separable Training (SALT) under the NTK framework. Figure 6 (a) shows that SALT achieves lower loss values and converges faster, suggesting more efficient optimization. Figure 6 (b) illustrates the eigenvalue distribution: SINGLE's eigenvalues are heavily concentrated near zero, indicating limited exploration of the feature space, whereas SALT maintains a broader spectrum that facilitates richer representation learning.

To formalize the NTK Frobenius norm, we define the NTK matrix at training iteration $t$ as $\Theta_t \in \mathbb{R}^{n \times n}$, where $n$ is the number of training samples. The Frobenius norm of $\Theta_t$ is given by:

$$\|\Theta_t\|_F = \sqrt{\sum_{i=1}^{n} \sum_{j=1}^{n} \Theta_{t,ij}^2}. \tag{45}$$

Since the NTK matrix $\Theta_t$ is symmetric, its Frobenius norm can also be computed using its eigenvalues $\lambda_i(\Theta_t)$:

$$\|\Theta_t\|_F = \sqrt{\sum_{i=1}^{n} \lambda_i^2(\Theta_t)}. \tag{46}$$

Figure 6 (c) shows that the NTK Frobenius norm over training remains both higher and more stable for SALT, whereas SINGLE exhibits lower norms with abrupt fluctuations, implying less structured gradient propagation.

To analyze the stability of the training dynamics, we define the variation in Frobenius norm as the difference between consecutive iterations:

$$\Delta\|\Theta_t\|_F = \|\Theta_t\|_F - \|\Theta_{t-1}\|_F. \tag{47}$$

Figure 6 (d) illustrates this variation over training. A larger magnitude of $\Delta\|\Theta_t\|_F$ indicates abrupt changes in the optimization process, while smaller variations suggest a more stable learning trajectory. SINGLE undergoes sporadic jumps, whereas SALT displays smoother transitions:

$$\text{Variance}(\Delta\|\Theta_t\|_F) = \mathbb{E}[(\Delta\|\Theta_t\|_F)^2] - (\mathbb{E}[\Delta\|\Theta_t\|_F])^2. \tag{48}$$

A lower variance in $\Delta\|\Theta_t\|_F$ implies more consistent gradient updates, facilitating improved generalization. Taken together, these results demonstrate that Separable Training not only accelerates convergence but also promotes more robust and comprehensive feature extraction, leading to stronger generalization than SINGLE.

# D ALGORITHMS

# E MOTIVATION

## E.1 SPECTRAL BIAS AND OPTIMIZATION CHALLENGES

Neural networks are known to exhibit certain biases and pathologies during training, especially when learning from complex or heterogeneous signals. One well-studied phenomenon is **spectral**

---

**Algorithm 2:** SALT

---

**Input:** Input time-series data $X_{\text{in}}$, Any forecasting model $f_\theta^{(i)}$ for each component $i$, MSE loss function $L$, maximum residual iteration number $max\_iter = N$, maximum train iteration number $train\_iter = M$

1 $i \leftarrow 1$;

2 Decompose $X_{In}$ into $P^{(1)}$ and $U^{(1)}$;

3 Initialize model parameters $\theta_1$;

4 **while** $i \leq max\_iter$ **do**

5      Train model $f_\theta^{(i)}$ and parameter on component $P^{(i)}$ using:

$$\hat{P}_{out}^{(i)} = f_\theta^{(i)}(P^{(i)}) \tag{49}$$

        **while** $j \leq train\_iter$ **do**

6          Update model parameters $f_\theta^{(i)}$ using gradient descent with loss:

$$\theta_i := \theta_i - \eta \nabla_{\theta_i} L(P_{\text{out}}^{(i)}, \hat{P}_{\text{out}}^{(i)}) \tag{50}$$

         $j \leftarrow j + 1$;

7      Decompose $U^{(i)}$ into $P^{(i+1)}$ and $U^{(i+1)}$;

8      Initialize model parameters $\theta_{i+1}$;

9      $i \leftarrow i + 1$;

10 **return** final prediction $\hat{Y}_{\text{out}} = \sum_{i=1}^{N} \hat{P}_{\text{out}}^{(i)}$

---

**bias** (Rahaman et al., 2019; Cao et al., 2019), where models tend to learn low-frequency components—typically smooth, slowly varying signals—before higher-frequency ones. This tendency is especially pronounced in the early phases of training and is believed to arise from both architectural properties and gradient-based optimization dynamics.

This behavior has been theoretically supported by the Neural Tangent Kernel (NTK) framework (Jacot et al., 2018), which characterizes how infinitely wide networks evolve under gradient descent. NTK theory shows that networks prioritize directions in function space corresponding to larger eigenvalues of the NTK Gram matrix, which tend to align with low-complexity (i.e., low-frequency) functions (Valle-Perez et al., 2018). As a result, higher-frequency or more complex structures remain underrepresented, especially when the training signal contains overlapping components of varying predictability.

Another critical challenge is **gradient interference**, which arises when gradients from structurally distinct components — such as trend, seasonality, and residual noise — conflict during optimization. This has been observed in both multi-task learning and time-series settings (Chai et al., 2022), where signals with differing temporal or frequency characteristics are trained using a single global loss function. Such interference can lead to unstable convergence, biased updates, or suboptimal model capacity allocation.

In time-series forecasting, these challenges are especially salient due to the inherently structured and heterogeneous nature of the data. A single time-series can simultaneously exhibit long-term trends, periodic cycles, and stochastic noise, all of which interact during optimization. When these components are jointly optimized using a single-loss, low-frequency patterns tend to dominate early learning, while high-frequency components are either ignored or learned poorly. These optimization behaviors not only slow convergence but also limit generalization, particularly under distribution shifts or non-stationary dynamics.

**Empirical Evidence in Time-Series Forecasting:** To explore how these theoretical insights manifest in practice, we analyze learning behavior across decomposed time-series patterns. Figure 3 (Top) shows cosine similarity matrices between gradients of different components, revealing frequent conflicts and negative alignment between structurally distinct patterns—particularly between dominant and residual components. Figure 3 (Bottom) further demonstrates that simpler patterns (e.g., low-frequency components) consistently achieve faster loss reduction in early epochs. These observations are strongly aligned with theoretical predictions from spectral bias and gradient interference literature, and underscore the need for training strategies that explicitly address these chal-

lenges. This trend has also been empirically validated in recent work (Ackaah-Gyasi et al., 2023), which shows that spectral bias consistently emerges in Transformer-based long sequence forecasting models on both synthetic and real-world time-series datasets.

### E.2 COMPARISON WITH EXISTING APPROACHES

In this section, we provide a comprehensive comparison between the SALT framework and existing approaches that share some similarities in concept but differ in execution and objective. This comparison aims to highlight the unique aspects of the SALT methodology, particularly the combination of iterative residual decomposition and Separable Training.

**Overview of existing residual-based methods:** Residual-based methods, such as Boosting Algorithms (e.g., Gradient Boosting, AdaBoost), iteratively learn from residuals by refining predictions based on the errors of the previous models. These methods share a conceptual similarity with our iterative residual decomposition approach, as both aim to minimize residual errors over successive iterations. However, unlike iterative residual decomposition, which iteratively decomposes time-series data to extract meaningful patterns, boosting algorithms primarily focus on reducing residuals within machine learning models and do not perform decomposition iteratively on the time-series itself.

**Separable Training and ensemble learning approaches:** Ensemble learning methods, such as Bagging and Stacking, train different models on various subsets or aspects of the data and then aggregate their predictions. While this shares the notion of training models separately, it fundamentally differs from SALT's Separable Training, which explicitly decomposes and trains individual time-series components (e.g., trend, seasonality, residuals) independently. This ensures that each component is learned in a focused manner, unlike ensemble methods that do not distinguish between the underlying structures of the data.

**Recent time-series decomposition models:** Recent models like DLinear and Autoformer have incorporated decomposition techniques to better handle the inherent complexity of time-series data. These models often decompose time-series into different components for better prediction accuracy, similar to our decomposition process. However, unlike the iterative nature of the iterative residual decomposition in SALT, where decomposition is repeatedly applied to refine residuals until they converge towards white noise, these models typically apply decomposition only once. This iterative refinement sets SALT apart in its ability to extract and reduce overlooked predictable patterns more effectively.

**Hybrid approaches in time-series forecasting:** Hybrid models combining ARIMA with deep learning models aim to leverage the strengths of both traditional statistical methods and modern machine learning techniques for time-series forecasting. While they handle different aspects of the data, they do not achieve the systematic separation and independent training of components as in SALT's Separable Training. The iterative residual decomposition and Separable Training combination provides a more structured way of addressing noise and patterns than simply combining model outputs.

In summary, while there exist approaches that share certain aspects with SALT, such as residual learning, ensemble learning, and decomposition-based methods, none of them achieve the same level of effectiveness in addressing the bias-variance tradeoff in time-series forecasting. The iterative residual decomposition provides an unbiased, systematic extraction of predictable patterns, while Separable Training ensures that each component is learned without interference from others, leading to improved generalization and forecasting accuracy. This combination is unique to SALT and represents a significant advancement in handling the complexity and non-stationarity inherent in real-world time-series data.

## F DECOMPOSITION METHOD SELECTION

We design a lightweight feature extraction routine to automatically determine which base decomposition method to apply for each dataset. For a given time-series, we compute the following features:

- Autocorrelation: ACF at fixed lag (e.g., 24 or 144)

- Smoothness: Variance of first-order differences

- High-frequency Energy: Proportion of power above a threshold from the periodogram

- Stationarity: p-value from Augmented Dickey–Fuller (ADF) test

These features are mapped to base decomposition methods according to the rule table below: Based

Table 2: Statistical criteria for decomposition method selection.

| Condition | Selected Method |
|---|---|
| ACF > 0.7 | STL (strong seasonality) |
| Diff. Var < 0.05 | Moving Average (smooth signal) |
| High-freq energy > 0.1 | Frequency filter (noisy signal) |
| Otherwise | STR (default short-term residual) |

on these statistics, we assign each variable to one of the candidate decomposition strategies. Then, following a majority voting scheme across all variables in a dataset, we determine the representative decomposition method for that dataset. In cases where multiple strategies are comparably supported, we select the method that yields better empirical forecasting performance. Table 2 summarizes the feature statistics and the selected decomposition strategy for each dataset. We list all statistic results in Table 3 - 10

Table 3: Statistical criteria for Weather

| Variable | ACF | Diff.Var | High-freq energy | ADF p-value |
|---|---|---|---|---|
| p (mbar) | 0.984 | 0.01008 | 0.000 | 0.000 |
| T (degC) | 0.898 | 0.06433 | 0.000 | 0.000 |
| Tpot (K) | 0.899 | 0.06551 | 0.000 | 0.000 |
| Tdew (degC) | 0.965 | 0.04196 | 0.000 | 0.000 |
| rh (%) | 0.727 | 1.80234 | 0.000 | 0.000 |
| VPmax (mbar) | 0.870 | 0.09052 | 0.000 | 0.000 |
| VPact (mbar) | 0.963 | 0.02346 | 0.000 | 0.000 |
| VPdef (mbar) | 0.724 | 0.12111 | 0.000 | 0.000 |
| sh (g/kg) | 0.963 | 0.00948 | 0.000 | 0.000 |
| H2OC (mmol/mol) | 0.963 | 0.02396 | 0.000 | 0.000 |
| rho (g/m$^3$) | 0.917 | 1.19933 | 0.000 | 0.000 |
| wv (m/s) | 0.001 | 3796.89623 | 0.400 | 0.000 |
| max. wv (m/s) | 0.558 | 0.87276 | 0.023 | 0.000 |
| wd (deg) | 0.360 | 4295.56845 | 0.113 | 0.000 |
| rain (mm) | 0.082 | 0.00957 | 0.088 | 0.000 |
| raining (s) | 0.298 | 3490.50954 | 0.038 | 0.000 |
| SWDR (W/m$^2$) | 0.440 | 3471.93023 | 0.010 | 0.000 |
| PAR (μmol/m$^2$/s) | 0.443 | 12594.55129 | 0.010 | 0.000 |
| max. PAR (μmol/m$^2$/s) | 0.330 | 118352.01386 | 0.063 | 0.000 |
| Tlog (degC) | 0.876 | 0.04368 | 0.000 | 0.000 |
| OT | 0.202 | 8194.52490 | 0.009 | 0.000 |

Table 4: Statistical criteria for Exchange Rate

| Variable | ACF | Diff.Var | High-freq energy | ADF p-value |
|---|---|---|---|---|
| 0 | 0.982 | 0.00004 | 0.000 | 0.449 |
| 1 | 0.946 | 0.00010 | 0.001 | 0.225 |
| 2 | 0.987 | 0.00002 | 0.000 | 0.605 |
| 3 | 0.983 | 0.00004 | 0.000 | 0.490 |
| 4 | 0.975 | 0.00000 | 0.001 | 0.049 |
| 5 | 0.974 | 0.00000 | 0.000 | 0.237 |
| 6 | 0.987 | 0.00001 | 0.000 | 0.407 |
| OT | 0.982 | 0.00002 | 0.000 | 0.417 |

Table 5: Statistical criteria for ETTh1

| Variable | ACF | Diff.Var | High-freq energy | ADF p-value |
|----------|-----|----------|------------------|-------------|
| HUFL | 0.792 | 8.48856 | 0.014 | 0.000 |
| HULL | 0.816 | 0.66820 | 0.023 | 0.000 |
| MUFL | 0.794 | 7.64395 | 0.013 | 0.000 |
| MULL | 0.827 | 0.53333 | 0.024 | 0.000 |
| LUFL | 0.604 | 0.32911 | 0.027 | 0.000 |
| LULL | 0.821 | 0.03153 | 0.013 | 0.000 |
| OT | 0.940 | 0.84306 | 0.001 | 0.008 |

Table 6: Statistical criteria for ETTm1

| Variable | ACF | Diff.Var | High-freq energy | ADF p-value |
|----------|-----|----------|------------------|-------------|
| HUFL | 0.120 | 2.41251 | 0.007 | 0.000 |
| HULL | 0.665 | 0.32160 | 0.013 | 0.000 |
| MUFL | 0.107 | 2.26509 | 0.007 | 0.000 |
| MULL | 0.679 | 0.25678 | 0.013 | 0.000 |
| LUFL | 0.511 | 0.08376 | 0.007 | 0.000 |
| LULL | 0.861 | 0.01439 | 0.007 | 0.000 |
| OT | 0.966 | 0.19841 | 0.000 | 0.001 |

## G EXPERIMENTS

### G.1 ABLATION STUDIES

In Table 11, we evaluate different decomposition strategies used in SALT across multiple base models. SALT consistently improves forecasting performance across all architectures and datasets, regardless of the decomposition method used (STR, STL, Frequency-based), validating its robustness to decomposition choices.

In Table 12, we assess the effect of input noise on performance to ensure fairness in comparison. Even when SINGLE models are trained with denoised inputs, they fail to outperform SALT. This confirms that the performance gains of SALT are not due to differences in input quality but due to its structural advantages.

**Comparison between SALT and Boosting-based Ensembles:** To distinguish the effectiveness of our method (SALT) from conventional ensembling strategies, we compare it against two baselines—using the same TSF backbone, hyperparameters, and data splits: a boosting-style ensemble (sequentially trained for $K = 5$ rounds to fit residuals) and a standard ensemble of five independent runs ($M = 5$). As shown in Table 13, while both ensemble methods provide only minor improvements (0.5 pp over the base TSF), SALT consistently achieves the best performance (1–2 pp gain) across all settings on ETTh1 and ETTh2 datasets. These results confirm that the performance gains of SALT are not merely due to ensembling effects, but stem from its structure-aligned training strategy. This advantage stems from SALT's structure-wise parallel training, which contrasts with boosting's sequential residual correction. Whereas boosting incrementally adjusts predictions to reduce error, SALT isolates predictable components and trains dedicated models for each—improving specialization, interpretability, and robustness.

### G.2 EXPERIMENTAL RESULTS

In Table 14 and Table 15, we report the full forecasting performance of SALT across various base models and benchmark dataset.

### G.3 RESULTS WITH DIFFERENT RANDOM SEEDS

To examine the robustness of our results, we train our SALT model with 3 different random seeds. The mean and standard deviation of the results are reported in Table 16

Table 7: Statistical criteria for ETTh2

| Variable | ACF | Diff.Var | High-freq energy | ADF p-value |
|----------|------|----------|------------------|-------------|
| HUFL | 0.774 | 10.46711 | 0.015 | 0.000 |
| HULL | 0.856 | 3.27086 | 0.016 | 0.000 |
| MUFL | 0.905 | 8.23228 | 0.007 | 0.001 |
| MULL | 0.852 | 2.05857 | 0.019 | 0.000 |
| LUFL | 0.959 | 1.16871 | 0.006 | 0.132 |
| LULL | 0.959 | 0.37738 | 0.002 | 0.013 |
| OT | 0.933 | 1.67725 | 0.000 | 0.006 |

Table 8: Statistical criteria for ETTm2

| Variable | ACF | Diff.Var | High-freq energy | ADF p-value |
|----------|------|----------|------------------|-------------|
| HUFL | 0.819 | 4.10093 | 0.005 | 0.000 |
| HULL | 0.905 | 1.34238 | 0.005 | 0.000 |
| MUFL | 0.910 | 3.00546 | 0.002 | 0.000 |
| MULL | 0.895 | 0.71561 | 0.005 | 0.000 |
| LUFL | 0.964 | 0.69124 | 0.003 | 0.013 |
| LULL | 0.983 | 0.17837 | 0.001 | 0.007 |
| OT | 0.887 | 0.16472 | 0.000 | 0.001 |

## G.4 EXPERIMENTAL SETTINGS

**Environment Setting:** We conduct experiments on multivariate time-series forecasting. All experiments were conducted in the same software and hardware environments. UBUNTU 18.04 LTS, PYTHON 3.8.0, NUMPY 1.22.3, SCIPY 1.10.1, MATPLOTLIB 3.6.2, PYTORCH 2.0.1, CUDA 11.4, NVIDIA Driver 470.182.03 i9 CPU, and NVIDIA RTX A5000.

## G.5 DATASETS

We conducted experiments on 9 benchmarked LTSF datasets, including Weather, Exchange rate, ETTh1, ETTh2, ETTm1, ETTm2, Electricity, Traffic, and National illness (ILL). Additionally, we conducted stationarity testing on each benchmarked dataset using time-series stationarity tests, including the Augmented Dickey-Fuller (ADF) test (Mushtaq, 2011) and the Kwiatkowski-Phillips-Schmidt-Shin (KPSS) test (Baum, 2018). The tests were performed for each variable, and the overall characteristics of stationarity were documented accordingly (detailed results of stationarity testing are in Appendix H.1). Based on the statistical test results, we classify the datasets into non-stationary (Weather, Exchange, ETTh1, ETTm1) and weak non-stationary (ETTh2, ETTm2, Electricity, Traffic, ILL) categories.

1. Weather dataset consists of measurements for 21 weather indicators, such as temperature and humidity, collected every 10 minutes throughout the year 2020 (Wu et al., 2021).

2. Exchange dataset includes exchange rate data among 8 different countries (Lai et al., 2018).

3. ETT (Electricity Transformer Temperature) comprises four datasets: two with hourly granularity and two with 15-minute granularity, recorded between July 2016 and July 2018. Each dataset contains seven features related to oil and load conditions of transformers (Zhou et al., 2021).

4. Electricity dataset tracks the hourly electricity consumption of 321 clients from 2012 to 2014.

5. Traffic dataset represents the road occupancy rates, capturing hourly data recorded by sensors on the San Francisco freeways between 2015 and 2016 (Wu et al., 2021).

6. ILL (Influenza-like Illness) dataset is provided by the Centers for Disease Control and Prevention (CDC) of the United States, covering the period from 2002 to 2021 (Wu et al., 2021).

**Dataset Sources:** The datasets used in our experiments are publicly available and can be downloaded at the following locations:

Table 9: Statistical criteria for Electricity

| Variable | ACF | Diff.Var | High-freq energy | ADF p-value |
|----------|------|-------------|-----------------|-------------|
| 0 | 0.752 | 155.87009 | 0.040 | 0.000 |
| 1 | 0.776 | 170.42516 | 0.018 | 0.000 |
| 2 | 0.626 | 536.36924 | 0.020 | 0.000 |
| 3 | 0.895 | 5219.13626 | 0.012 | 0.000 |
| 4 | 0.876 | 802.27979 | 0.013 | 0.000 |
| 5 | 0.906 | 13799.12173 | 0.012 | 0.000 |
| 6 | 0.898 | 19.33540 | 0.003 | 0.000 |
| 7 | 0.848 | 14725.25833 | 0.020 | 0.000 |
| 8 | 0.739 | 2638.22678 | 0.028 | 0.000 |
| 9 | 0.564 | 2839.85192 | 0.026 | 0.000 |
| 10 | 0.875 | 532.62733 | 0.017 | 0.000 |
| 11 | 0.661 | 4817.97673 | 0.014 | 0.000 |
| 12 | 0.770 | 593.03945 | 0.006 | 0.000 |
| 13 | 0.869 | 486.69065 | 0.012 | 0.000 |
| 14 | 0.904 | 681.96855 | 0.017 | 0.000 |
| 15 | 0.832 | 26390.51993 | 0.016 | 0.000 |
| 16 | 0.809 | 33.79474 | 0.032 | 0.000 |
| 17 | 0.775 | 515.00245 | 0.023 | 0.000 |
| 18 | 0.889 | 7926.32310 | 0.011 | 0.000 |
| 19 | 0.724 | 940.50553 | 0.026 | 0.000 |

Table 10: Statistical criteria for Traffic

| Variable | ACF | Diff.Var | High-freq energy | ADF p-value |
|----------|------|---------|-----------------|-------------|
| 0 | 0.701 | 0.00208 | 0.034 | 0.000 |
| 1 | 0.718 | 0.00095 | 0.036 | 0.000 |
| 2 | 0.725 | 0.00189 | 0.025 | 0.000 |
| 3 | 0.668 | 0.00061 | 0.046 | 0.000 |
| 4 | 0.754 | 0.00044 | 0.017 | 0.000 |
| 5 | 0.779 | 0.00134 | 0.020 | 0.000 |
| 6 | 0.777 | 0.00107 | 0.016 | 0.000 |
| 7 | 0.835 | 0.00079 | 0.013 | 0.000 |
| 8 | 0.845 | 0.00033 | 0.011 | 0.000 |
| 9 | 0.672 | 0.00557 | 0.061 | 0.000 |
| 10 | 0.722 | 0.00063 | 0.046 | 0.000 |
| 11 | 0.727 | 0.00108 | 0.020 | 0.000 |
| 12 | 0.600 | 0.00183 | 0.063 | 0.000 |
| 13 | 0.804 | 0.00039 | 0.024 | 0.000 |
| 14 | 0.762 | 0.00135 | 0.023 | 0.000 |
| 15 | 0.650 | 0.00093 | 0.044 | 0.000 |
| 16 | 0.746 | 0.00019 | 0.014 | 0.000 |
| 17 | 0.760 | 0.00123 | 0.029 | 0.000 |
| 18 | 0.815 | 0.00016 | 0.013 | 0.000 |
| 19 | 0.738 | 0.00235 | 0.025 | 0.000 |

1. Weather:https://drive.google.com/drive/folders/1ZOYpTUa82_jCcxIdTmyr0LXQfvaM9vIy,

2. Exchange:https://drive.google.com/drive/folders/1ZOYpTUa82_jCcxIdTmyr0LXQfvaM9vIy,

3. ETT:https://drive.google.com/drive/folders/1ZOYpTUa82_jCcxIdTmyr0LXQfvaM9vIy

4. Electricity:https://archive.ics.uci.edu/dataset/321/electricityloaddiagrams20112014,

5. Traffic:https://pems.dot.ca.gov/,

6. ILI:https://drive.google.com/drive/folders/1ZOYpTUa82_jCcxIdTmyr0LXQfvaM9vIy

## G.6 BASELINES

We evaluated our model compared to the following state-of-the-art baselines:

1. N-BEATS (Oreshkin et al., 2019) is a deep learning architecture specifically designed for univariate time-series forecasting. It is based on backward and forward residual links and a

Table 11: Ablation studies on various time-series decomposition methods with SALT.

| Decomposition method | ETTh1 | | | | | | ILL | | | | | |
|---|---|---|---|---|---|---|---|---|---|---|---|---|
| | N-BEATS | N-HiTS | DLinear | PatchTST | iTransformer | TimeMixer | N-BEATS | N-HiTS | DLinear | PatchTST | iTransformer | TimeMixer |
| Original model | 0.387 | 0.364 | 0.375 | 0.370 | 0.386 | 0.375 | **1.879** | 1.862 | 2.215 | 1.319 | 2.085 | 1.469 |
| SALT (STR) | 0.344 | 0.363 | 0.320 | 0.317 | 0.342 | 0.342 | 1.879 | 1.563 | 1.997 | 1.289 | 2.109 | 1.508 |
| SALT (STL) | **0.321** | **0.353** | **0.251** | **0.273** | **0.302** | **0.260** | 1.692 | **1.543** | **1.931** | **1.078** | **1.798** | **1.327** |
| SALT (Freq.) | 0.348 | 0.357 | 0.316 | 0.295 | 0.343 | 0.341 | 1.889 | 1.602 | 2.161 | 1.303 | 2.124 | 1.472 |

Table 12: Effect of input noise on model performance. Despite using denoised inputs, SINGLE does not outperform SALT, confirming fairness in comparison.

| Method (PatchTST) | ETTh1 | ETTh2 | ILL | Method (DLinear) | ETTh1 | ETTh2 | ILL |
|---|---|---|---|---|---|---|---|
| SINGLE (w noise) | 0.370 | 0.274 | 1.319 | SINGLE (w noise) | 0.375 | 0.289 | 2.215 |
| SINGLE (w/o noise) | 0.299 | 0.282 | 2.582 | SINGLE (w/o noise) | 0.325 | 0.283 | 2.464 |
| SALT (w noise) | 0.288 | 0.278 | 1.618 | SALT (w noise) | 0.267 | 0.281 | 2.072 |
| SALT (w/o noise) | **0.273** | **0.263** | **1.078** | SALT (w/o noise) | **0.251** | **0.273** | **1.931** |

deep stack of fully-connected layers. Each block in N-BEATS is trained to produce a forecast and a backcast (residual of the input), and subsequent blocks refine the residuals. The model is fully interpretable when using trend and seasonality basis functions and achieves strong empirical performance without relying on recurrence or convolutions.

2. N-HiTS (Challu et al., 2023) is an extension of the N-BEATS architecture, designed to improve scalability and accuracy for long-horizon forecasting. It introduces three major innovations: hierarchical forecasting blocks that operate at multiple temporal resolutions; interpolation-based skip connections to avoid redundant computation; and block specialization, where different blocks learn different frequency patterns. These changes allow N-HiTS to achieve better performance and computational efficiency, especially on high-resolution and long-range forecasting tasks.

3. DLinear (Zeng et al., 2023) (Decomposition Linear) paper investigates the effectiveness of Transformer-based models for long-term time-series forecasting (LTSF). The authors challenge the dominance of Transformers by introducing a simple one-layer linear model, LTSF-Linear, which surprisingly outperforms Transformer-based LTSF models on nine real-life datasets. They argue that Transformers may not be ideal for LTSF because of the temporal information loss caused by the permutation-invariant self-attention mechanism. Their experiments demonstrate that LTSF-Linear, with its simple structure and trend-seasonality decomposition, achieves superior performance compared to complex Transformer models, suggesting that simpler models could be more suitable for certain time-series forecasting tasks.

4. PatchTST (Nie et al., 2022) is a Transformer-based model designed for multivariate time-series forecasting and self-supervised representation learning. It introduces two key components: patching, where time-series data is segmented into subseries-level patches to serve as input tokens, enhancing local semantic information and reducing computation; and channel-independence, where each channel contains a univariate time-series that shares the same embedding and Transformer weights across all series. This approach allows PatchTST to handle longer look-back windows and capture essential temporal information, leading to significant improvements in long-term forecasting accuracy over other Transformer-based models, particularly when dealing with large datasets.

5. iTransformer (Liu et al., 2023) is a modified Transformer architecture designed specifically for time-series forecasting. Unlike traditional Transformers that embed multiple time-series variables as temporal tokens, iTransformer takes an inverted approach by embedding each time-series as variate tokens. This method allows the attention mechanism to capture multivariate correlations more effectively, while a feed-forward network learns nonlinear representations of individual time-series. By focusing on the relationships among variates, iTransformer achieves enhanced performance, generalization across different variates, and

Table 13: Comparison of boosting-based ensemble and our method (SALT).

| Method | ETTh1 ($h = 96$) | | | |
| --- | --- | --- | --- | --- |
| | DLinear | PatchTST | iTransformer | TimeMixer |
| Original model | 0.375 | 0.370 | 0.386 | 0.375 |
| Boosting (GBM-style) | 0.501 | 0.501 | 0.501 | 0.501 |
| Ensemble of 5 runs | 0.314 | 0.309 | 0.326 | 0.318 |
| SALT | **0.251** | **0.273** | **0.302** | **0.260** |

| Method | ETTh2 ($h = 96$) | | | |
| --- | --- | --- | --- | --- |
| | DLinear | PatchTST | iTransformer | TimeMixer |
| Original model | 0.289 | 0.274 | 0.297 | 0.289 |
| Boosting (GBM-style) | 0.498 | 0.498 | 0.498 | 0.498 |
| Ensemble of 5 runs | 0.275 | 0.282 | 0.302 | 0.281 |
| SALT | **0.273** | **0.263** | **0.293** | **0.278** |

Table 14: Experimental results on 9 benchmark datasets. Results where the SALT method improved the performance of a base model are highlighted in **bold**.

| Datasets | $h$ | Weather | | Exchange | | ETTh1 | | ETTm1 | | ETTh2 | | ETTm2 | | Electricity | | Traffic | | $h$ | ILL | |
| --- | --- | --- | --- | --- | --- | --- | --- | --- | --- | --- | --- | --- | --- | --- | --- | --- | --- | --- | --- | --- |
| | | MSE | MAE | MSE | MAE | MSE | MAE | MSE | MAE | MSE | MAE | MSE | MAE | MSE | MAE | MSE | MAE | | MSE | MAE |
| N-BEATS (2021) | 96 | 0.167 | 0.281 | 0.098 | 0.206 | 0.387 | 0.410 | 0.320 | 0.362 | 0.303 | 0.363 | 0.184 | 0.263 | 0.145 | 0.247 | 0.398 | 0.282 | 24 | 1.879 | 0.886 |
| | 192 | 0.229 | 0.333 | 0.225 | 0.329 | 0.428 | 0.434 | 0.353 | 0.379 | 0.364 | 0.402 | 0.246 | 0.307 | 0.180 | 0.283 | 0.409 | 0.293 | 36 | 2.210 | 1.018 |
| | 336 | 0.289 | 0.396 | 0.493 | 0.482 | 0.448 | 0.447 | 0.403 | 0.412 | 0.360 | 0.407 | 0.309 | 0.355 | 0.200 | 0.308 | 0.449 | 0.318 | 48 | 2.440 | 1.088 |
| | 720 | 0.368 | 0.459 | 1.108 | 0.804 | 0.466 | 0.471 | 0.458 | 0.442 | 0.428 | 0.465 | 0.411 | 0.425 | 0.266 | 0.362 | 0.589 | 0.391 | 60 | 2.547 | 1.057 |
| SALT N-BEATS | 96 | **0.165** | **0.206** | **0.085** | **0.201** | **0.321** | **0.376** | **0.253** | **0.312** | **0.283** | **0.336** | **0.177** | **0.259** | **0.144** | **0.244** | **0.396** | **0.280** | 24 | **1.692** | **0.872** |
| | 192 | **0.216** | **0.260** | **0.190** | **0.311** | **0.376** | **0.412** | **0.296** | **0.347** | **0.360** | **0.389** | **0.246** | **0.307** | **0.175** | **0.280** | **0.406** | **0.290** | 36 | **2.013** | **0.968** |
| | 336 | **0.265** | **0.298** | **0.399** | **0.461** | **0.405** | **0.435** | **0.328** | **0.369** | **0.356** | 0.411 | **0.306** | **0.350** | **0.195** | **0.302** | **0.432** | **0.307** | 48 | **2.001** | **0.987** |
| | 720 | **0.339** | **0.346** | **1.101** | **0.801** | **0.441** | **0.472** | **0.393** | **0.402** | **0.416** | **0.441** | **0.403** | **0.406** | **0.265** | **0.360** | **0.582** | **0.383** | 60 | **2.128** | **1.029** |
| Imp.(Avg.) | | 7.67% | 1.38% | 7.97% | 2.75% | 12.5% | 5.07% | 19.1% | 11.4% | 3.24% | 4.70% | 4.58% | 4.50% | 2.15% | 1.25% | 1.79% | 2.10% | | 11.7% | 4.16% |
| N-HiTS (2024) | 96 | 0.158 | 0.195 | 0.092 | 0.202 | 0.364 | 0.408 | 0.308 | 0.345 | 0.262 | 0.349 | 0.176 | 0.255 | 0.147 | 0.249 | 0.402 | 0.282 | 24 | 1.862 | 0.869 |
| | 192 | 0.211 | 0.247 | 0.208 | 0.322 | 0.406 | 0.412 | 0.346 | 0.361 | 0.341 | 0.388 | 0.245 | 0.305 | 0.167 | 0.269 | 0.420 | 0.297 | 36 | 2.071 | 0.934 |
| | 336 | 0.274 | 0.300 | 0.301 | 0.403 | 0.426 | 0.426 | 0.381 | 0.408 | 0.338 | 0.385 | 0.295 | 0.346 | 0.186 | 0.290 | 0.448 | 0.313 | 48 | 2.134 | 0.932 |
| | 720 | 0.351 | 0.353 | 0.798 | 0.596 | 0.447 | 0.459 | 0.437 | 0.419 | 0.407 | 0.448 | 0.401 | 0.413 | 0.243 | 0.340 | 0.539 | 0.353 | 60 | 2.137 | 0.968 |
| SALT N-HiTS | 96 | **0.138** | **0.179** | **0.090** | **0.192** | **0.354** | **0.389** | **0.288** | **0.335** | **0.258** | **0.342** | **0.168** | **0.246** | **0.140** | **0.225** | **0.388** | **0.277** | 24 | **1.543** | **0.793** |
| | 192 | **0.209** | **0.239** | **0.189** | **0.311** | **0.388** | **0.417** | **0.329** | **0.358** | **0.339** | **0.375** | **0.234** | **0.288** | **0.160** | **0.260** | **0.404** | **0.289** | 36 | **1.973** | **0.889** |
| | 336 | **0.257** | **0.298** | **0.297** | **0.389** | **0.419** | **0.435** | **0.366** | **0.402** | **0.336** | **0.367** | **0.289** | **0.316** | **0.178** | **0.288** | **0.415** | **0.304** | 48 | **2.130** | **0.915** |
| | 720 | **0.338** | **0.320** | **0.690** | **0.563** | **0.424** | **0.448** | **0.419** | **0.408** | **0.403** | **0.408** | **0.387** | **0.405** | **0.215** | **0.335** | **0.507** | **0.345** | 60 | **2.023** | **0.959** |
| Imp.(Avg.) | | 5.73% | 5.57% | 10.3% | 4.79% | 3.95% | 1.58% | 5.09% | 3.13% | 0.96% | 3.50% | 4.21% | 5.46% | 7.54% | 4.36% | 5.64% | 2.81% | | 6.86% | 4.43% |
| DLinear (2023) | 96 | 0.176 | 0.237 | 0.081 | 0.203 | 0.375 | 0.399 | 0.299 | 0.343 | 0.289 | 0.353 | 0.167 | 0.260 | 0.140 | 0.237 | 0.410 | 0.282 | 24 | 2.215 | 1.081 |
| | 192 | 0.220 | 0.282 | 0.157 | 0.293 | 0.405 | 0.416 | 0.335 | 0.365 | 0.383 | 0.418 | 0.224 | 0.303 | 0.153 | 0.249 | 0.423 | 0.287 | 36 | 1.963 | 0.963 |
| | 336 | 0.265 | 0.319 | 0.305 | 0.414 | 0.439 | 0.443 | 0.369 | 0.386 | 0.448 | 0.465 | 0.281 | 0.342 | 0.169 | 0.267 | 0.436 | 0.296 | 48 | 2.130 | 1.024 |
| | 720 | 0.323 | 0.362 | 0.643 | 0.601 | 0.472 | 0.490 | 0.425 | 0.421 | 0.605 | 0.551 | 0.397 | 0.421 | 0.203 | 0.301 | 0.466 | 0.315 | 60 | 2.368 | 1.096 |
| SALT DLinear | 96 | **0.146** | **0.208** | **0.067** | **0.174** | **0.251** | **0.336** | **0.250** | **0.312** | **0.274** | **0.338** | **0.146** | **0.243** | **0.125** | **0.230** | **0.340** | **0.273** | 24 | **1.931** | **0.945** |
| | 192 | **0.203** | **0.267** | **0.157** | **0.278** | **0.317** | **0.381** | **0.285** | **0.340** | **0.347** | **0.397** | **0.221** | **0.300** | **0.145** | **0.248** | **0.399** | **0.285** | 36 | **1.433** | **0.823** |
| | 336 | **0.257** | **0.317** | **0.303** | **0.412** | **0.361** | **0.423** | **0.319** | **0.363** | **0.429** | **0.453** | **0.273** | **0.337** | **0.160** | **0.265** | **0.266** | **0.283** | 48 | **1.512** | **0.832** |
| | 720 | **0.321** | 0.369 | **0.634** | 0.603 | **0.387** | **0.450** | **0.407** | **0.419** | **0.603** | **0.542** | **0.391** | **0.418** | **0.195** | **0.298** | **0.312** | **0.303** | 60 | **1.602** | **0.878** |
| Imp.(Avg.) | | 8.25% | 4.66% | 2.66% | -0.24% | 23.8% | 9.81% | 13.1% | 5.53% | 5.20% | 2.87% | 4.58% | 2.13% | 7.57% | 2.15% | 2.15% | -2.63% | | 29.1% | 16.9% |

improved handling of longer lookback windows, making it highly effective for complex multivariate time-series forecasting tasks.

6. **TimeMixer** (Wang et al., 2024) is a neural network architecture designed for long-term time-series forecasting. It employs a hierarchical design that captures temporal dependencies at multiple scales, efficiently modeling both short-term and long-term dependencies in time-series data. The architecture consists of multiple layers of specialized blocks, each handling different temporal scales, which allows the model to capture complex patterns across varying time intervals. By combining these blocks, TimeMixer achieves state-of-the-art performance on benchmark datasets, demonstrating its ability to handle diverse time-series forecasting tasks effectively.

**Baseline Sources:** The datasets used in our experiments are publicly available and can be downloaded at the following locations:

Table 15: Experimental results on 9 benchmark datasets. Results where the SALT method improved the performance of a base model are highlighted in **bold**.

| Datasets | h | Weather MSE | Weather MAE | Exchange MSE | Exchange MAE | ETTh1 MSE | ETTh1 MAE | ETTm1 MSE | ETTm1 MAE | ETTh2 MSE | ETTh2 MAE | ETTm2 MSE | ETTm2 MAE | Electricity MSE | Electricity MAE | Traffic MSE | Traffic MAE | h | ILL MSE | ILL MAE |
|---|---|---|---|---|---|---|---|---|---|---|---|---|---|---|---|---|---|---|---|---|
| PatchTST (2023) | 96 | 0.149 | 0.198 | 0.093 | 0.218 | 0.370 | 0.400 | 0.290 | 0.342 | 0.274 | 0.336 | 0.165 | 0.255 | 0.129 | 0.222 | 0.360 | 0.249 | 24 | 1.319 | 0.754 |
| | 192 | 0.194 | 0.241 | 0.208 | 0.332 | 0.413 | 0.429 | 0.332 | 0.369 | 0.339 | 0.379 | 0.220 | 0.292 | 0.147 | 0.240 | 0.379 | 0.256 | 36 | 1.007 | 0.870 |
| | 336 | 0.245 | 0.282 | 0.359 | 0.440 | 0.422 | 0.440 | 0.366 | 0.392 | 0.329 | 0.384 | 0.274 | 0.329 | 0.163 | 0.259 | 0.392 | 0.264 | 48 | 1.553 | 0.815 |
| | 720 | 0.314 | 0.334 | 1.194 | 0.815 | 0.447 | 0.468 | 0.416 | 0.420 | 0.379 | 0.422 | 0.362 | 0.385 | 0.197 | 0.290 | 0.432 | 0.286 | 60 | 1.016 | 0.788 |
| SALT PatchTST | 96 | **0.129** | **0.184** | **0.068** | **0.177** | **0.273** | **0.344** | **0.250** | **0.324** | **0.263** | **0.324** | **0.164** | **0.253** | **0.117** | **0.219** | **0.359** | **0.245** | 24 | **1.078** | **0.712** |
| | 192 | **0.180** | **0.226** | **0.166** | **0.288** | **0.306** | **0.369** | **0.287** | **0.340** | 0.338 | **0.375** | 0.225 | **0.290** | **0.143** | **0.238** | **0.376** | **0.252** | 36 | **1.128** | **0.738** |
| | 336 | **0.235** | **0.275** | 0.329 | 0.478 | **0.337** | **0.403** | **0.310** | **0.360** | 0.328 | **0.379** | 0.273 | **0.326** | **0.158** | **0.250** | **0.391** | **0.261** | 48 | **1.283** | **0.775** |
| | 720 | **0.304** | **0.327** | 0.877 | 0.698 | **0.351** | **0.419** | **0.356** | **0.389** | **0.378** | **0.419** | 0.360 | **0.383** | **0.194** | **0.288** | **0.431** | **0.281** | 60 | **1.210** | **0.763** |
| Imp.(Avg.) | | 7.64% | 5.15% | 21.3% | 13.3% | 24.6% | 12.2% | 15.8% | 8.31% | 1.57% | 1.99% | -0.22% | 0.35% | 4.79% | 2.41% | 0.21% | 0.31% | | 0.20% | 8.62% |
| iTransformer (2024) | 96 | 0.174 | 0.214 | 0.086 | 0.206 | 0.386 | 0.405 | 0.334 | 0.368 | 0.297 | 0.349 | 0.180 | 0.264 | 0.148 | 0.240 | 0.395 | 0.268 | 24 | 2.085 | 0.953 |
| | 192 | 0.221 | 0.254 | 0.177 | 0.299 | 0.441 | 0.436 | 0.377 | 0.391 | 0.380 | 0.400 | 0.250 | 0.309 | 0.162 | 0.253 | 0.417 | 0.276 | 36 | 1.973 | 0.947 |
| | 336 | 0.278 | 0.296 | 0.331 | 0.417 | 0.487 | 0.458 | 0.426 | 0.420 | 0.428 | 0.432 | 0.311 | 0.348 | 0.178 | 0.269 | 0.433 | 0.283 | 48 | 2.124 | 1.018 |
| | 720 | 0.358 | 0.347 | 0.847 | 0.691 | 0.503 | 0.491 | 0.491 | 0.459 | 0.427 | 0.445 | 0.412 | 0.407 | 0.225 | 0.317 | 0.467 | 0.302 | 60 | 2.164 | 1.032 |
| SALT iTransformer | 96 | **0.141** | **0.190** | **0.073** | **0.186** | **0.302** | **0.362** | **0.272** | **0.343** | **0.293** | **0.342** | **0.178** | **0.262** | **0.143** | **0.232** | **0.318** | **0.263** | 24 | **1.798** | **0.920** |
| | 192 | **0.196** | **0.240** | **0.170** | **0.288** | **0.330** | **0.390** | **0.328** | **0.383** | **0.374** | **0.392** | **0.246** | **0.303** | **0.158** | **0.250** | **0.385** | **0.273** | 36 | **1.550** | **0.870** |
| | 336 | **0.250** | **0.288** | **0.328** | **0.413** | **0.370** | **0.430** | **0.350** | **0.398** | **0.410** | **0.422** | **0.307** | **0.343** | **0.171** | **0.263** | **0.412** | **0.278** | 48 | **1.583** | **0.889** |
| | 720 | **0.331** | **0.340** | **0.730** | **0.649** | **0.402** | **0.458** | **0.393** | **0.420** | **0.422** | **0.434** | **0.410** | **0.404** | **0.205** | **0.303** | **0.456** | **0.308** | 60 | **1.603** | **0.877** |
| Imp.(Avg.) | | 12.7% | 6.66% | 9.64% | 5.84% | 24.1% | 9.54% | 18.0% | 6.07% | 1.42% | 0.23% | 0.08% | 0.212% | 4.27% | 0.09% | 8.07% | 0.54% | | 21.9% | 10.9% |
| TimeMixer (2024) | 96 | 0.163 | 0.209 | 0.093 | 0.212 | 0.375 | 0.400 | 0.320 | 0.357 | 0.289 | 0.341 | 0.175 | 0.258 | 0.153 | 0.247 | 0.462 | 0.285 | 24 | 1.469 | 0.798 |
| | 192 | 0.208 | 0.250 | 0.174 | 0.297 | 0.429 | 0.421 | 0.361 | 0.381 | 0.372 | 0.392 | 0.237 | 0.299 | 0.166 | 0.256 | 0.473 | 0.296 | 36 | 1.890 | 0.867 |
| | 336 | 0.251 | 0.287 | 0.349 | 0.426 | 0.484 | 0.458 | 0.390 | 0.404 | 0.386 | 0.414 | 0.298 | 0.340 | 0.185 | 0.277 | 0.498 | 0.296 | 48 | 1.885 | 0.924 |
| | 720 | 0.339 | 0.341 | 1.065 | 0.770 | 0.498 | 0.482 | 0.454 | 0.441 | 0.412 | 0.434 | 0.391 | 0.396 | 0.225 | 0.310 | 0.506 | 0.313 | 60 | 1.955 | 0.980 |
| SALT TimeMixer | 96 | **0.150** | **0.197** | **0.087** | **0.210** | **0.260** | **0.344** | **0.250** | **0.325** | **0.278** | **0.339** | **0.157** | **0.250** | **0.116** | **0.215** | **0.340** | **0.278** | 24 | **1.327** | **0.769** |
| | 192 | **0.203** | **0.243** | **0.170** | **0.294** | **0.331** | **0.389** | **0.290** | **0.350** | **0.348** | **0.386** | **0.234** | **0.289** | **0.153** | **0.250** | **0.401** | 0.289 | 36 | **1.073** | **0.703** |
| | 336 | **0.244** | **0.280** | **0.345** | **0.422** | **0.339** | **0.398** | **0.319** | **0.374** | **0.379** | **0.408** | **0.296** | **0.339** | **0.168** | **0.265** | **0.448** | 0.293 | 48 | **1.172** | **0.772** |
| | 720 | **0.335** | **0.338** | **0.950** | **0.731** | **0.386** | **0.438** | **0.375** | **0.407** | **0.410** | **0.428** | **0.383** | **0.389** | **0.211** | **0.308** | **0.441** | **0.308** | 60 | **1.523** | **0.838** |
| Imp.(Avg.) | | 4.53% | 3.35% | 4.98% | 1.43% | 27.3% | 12.2% | 20.3% | 8.62% | 3.35% | 0.10% | 4.24% | 0.88% | 12.4% | 6.15% | 17.5% | -3.09% | | 30.0% | 15.6% |

1. N-BEATS: https://github.com/philipperemy/n-beats,

2. N-HiTS: https://github.com/sktime/pytorch-forecasting,

3. DLinear: https://github.com/cure-lab/LTSF-Linear

4. PatchTST: https://github.com/yuqinie98/PatchTST,

5. iTransformer: https://github.com/thuml/iTransformer,

6. TimeMixer: https://github.com/kwuking/TimeMixer

## G.7 Hyperparameters

1. $h$ : forecasting horizon length

2. $I$ : input sequence length

3. $K$ : kernel size of time-series decomposition method

4. $p$ : period length of time-series decomposition method

5. $\lambda$ : learning rate

6. decomp.method : decomposition method

7. $e$ : training epochs

## G.8 Synthetic Experimental Settings

To demonstrate that SALT's training method learns more comprehensive information compared to single training under identical training times, we conducted synthetic experiments using a custom-generated time-series dataset. This experiment was specifically designed to validate the hypothesis that SALT can achieve richer learning by decomposing the dataset into distinct components and training them independently. The synthetic dataset includes multiple components, such as trend,

seasonality, and various types of noise (e.g., constant noise, uniform noise, and Gaussian noise), to simulate real-world complexity. The noise components were combined to create a challenging dataset that tests the robustness of each training approach.

In this experiment, the single training model was trained on the entire dataset for 30 epochs using a single-loss function. In contrast, SALT decomposed the dataset into three predictable patterns—trend and seasonality, and noise—and trained each component independently for 10 epochs, resulting in a total of 30 epochs across all components. By ensuring identical total training time, the comparison between single training and SALT remains fair and unbiased.

Using NTK-based analysis, we show that SALT effectively learns richer and more diverse information from the dataset compared to single training. The NTK eigenvalue distribution confirms that SALT captures higher diversity and complexity in the learning process, leading to better generalization and robustness. This result highlights SALT's ability to adapt to complex patterns in time-series data, particularly in the presence of challenging noise conditions.

### G.9 COMPUTATIONAL TIME AND MODEL USAGE

This section compares the time per epoch and memory usage between the original model and the proposed SALT framework during the training process (cf. Table 23 and Table 24). The proposed SALT applies Separable Training, where multiple stages of training are conducted. For instance, if the model undergoes 3 iterations, it learns 3 distinct predictable patterns separately. In this process, the time taken per epoch is similar to or shorter than that of the original model because the training data is processed in a more compact form than the raw data. **Furthermore, since each predictable pattern becomes simpler for the model to learn, fewer epochs are required for convergence at each stage. For example, if a model originally requires a total of 50 epochs for training, in Separable Training, each stage may converge within 10 to 15 epochs. Thus, even with training across 3 stages, only about 45 epochs in total are required, leading to a reduction in overall computation time.**

## H   ADDITIONAL STATIONARITIES

### H.1   STATIONARITY TEST ON BENCHMARKED DATASETS

In this section, we analyze whether each benchmarked dataset exhibits stationarity by applying two standard statistical tests: the Augmented Dickey–Fuller (ADF) test and the Kwiatkowski–Phillips–Schmidt–Shin (KPSS) test. Based on the results from Table 26 to Table 32, we summarize the overall stationarity characteristics in Table 25.

## I   VISUALIZATIONS

### I.1   SENSITIVITY ON SEPARABLE TRAINING ITERATIONS

In Figure 8, we show the sensitivity to iteration number $N$ of Separable Training on 9 datasets.

### I.2   VISUALIZATION OF PREDICTABLE PATTERNS

In Figure 9, we visualize the predictable patterns on 9 datasets.

### I.3   VISUALIZATION OF NTK GRAM MATRIX

In Figure 10, we visualize NTK's Gram matrix on real-world datasets. Additionally, from Figure 11 to Figure 13, we visualize NTK's Gram matrix on synthetic datasets (constant noise, Gaussian noise, and uniform noise). We observe that models trained with the conventional single-loss approach exhibit many regions in the Gram matrix with values close to zero (gray areas), indicating weak or absent gradient alignment. This suggests that certain interactions between features or components are not effectively captured during training. In contrast, the SALT yield Gram matrices with stronger

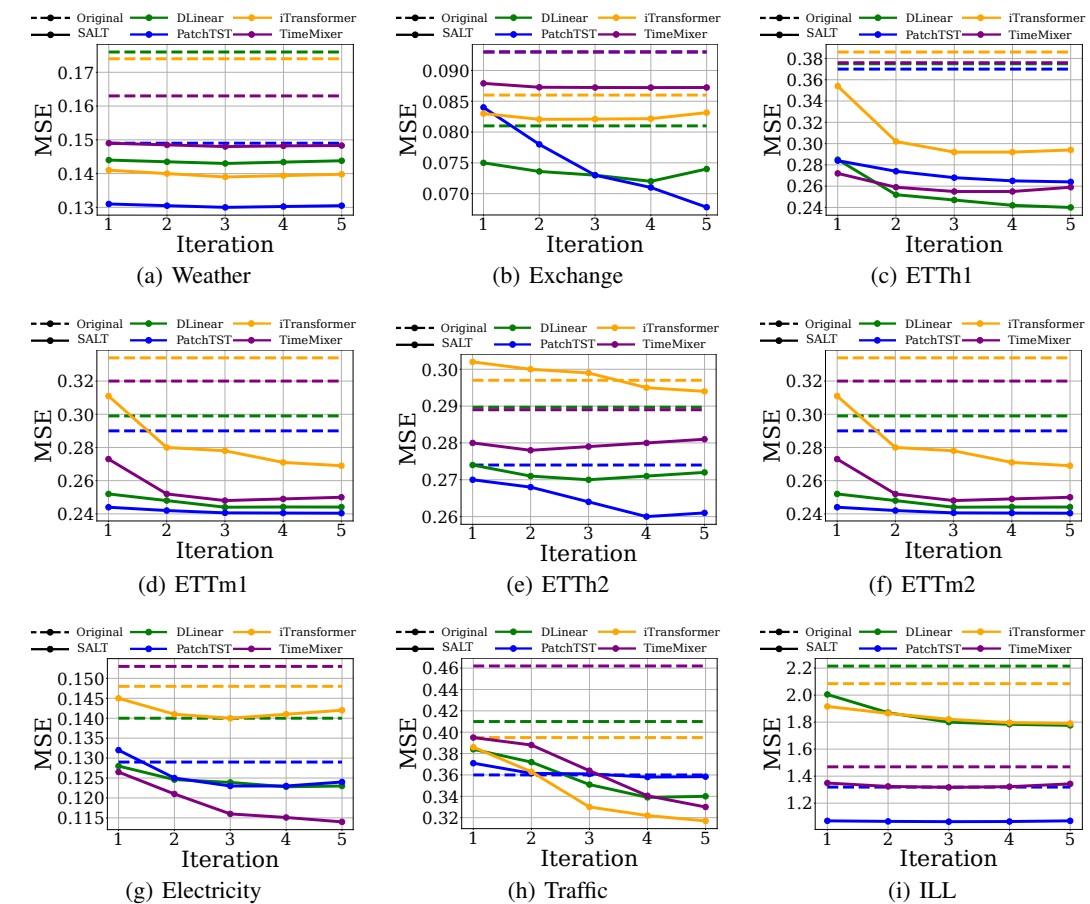

Figure 8: Sensitivity to iteration number $N$ of Separable Training.

### I.4 VISUALIZATION OF THE EMPIRICAL VALIDATION

To support the theoretical assumptions made in Appendix B, we empirically validate that the residuals produced by our Iterative Dominant Extraction (IDE) process converge toward white noise across real-world benchmark datasets.

First, Figure 14 visualizes the evolution of residual distributions across successive iterations of IDE. As the iteration progresses, the residuals (blue histograms) increasingly resemble a Gaussian distribution (red dashed line), suggesting that the STL-based decomposition progressively removes structured components such as trend and seasonality. This leads to residuals with more symmetric, bell-shaped distributions, indicative of zero-mean, homoskedastic behavior.

Second, Figure 16 empirically tests the i.i.d. white noise assumption through two standard statistical measures: the Durbin–Watson statistic (blue curve) and the mean autocorrelation (orange curve). Across all datasets, the Durbin–Watson statistic consistently approaches 2.0, while the autocorrelation converges toward 0.0 as the number of iterations increases. These trends confirm that temporal dependencies are effectively eliminated, and the residuals approximate i.i.d. noise.

Together, these results justify our modeling assumption in Theorem 1, where we treat the residual noise term $\zeta_j$ as an i.i.d. Gaussian variable.

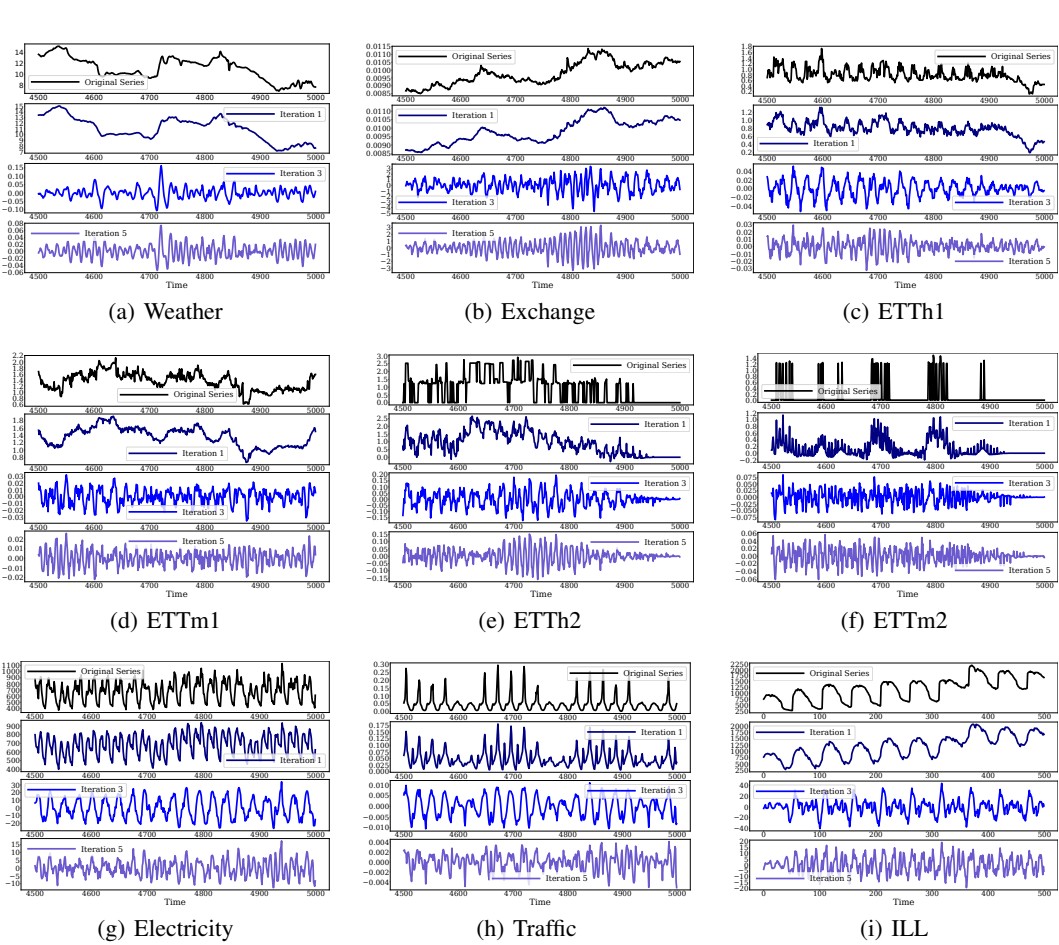

Figure 9: Visualization of the original time-series (top) and the predictable patterns at each iteration of the iterative residual decomposition.

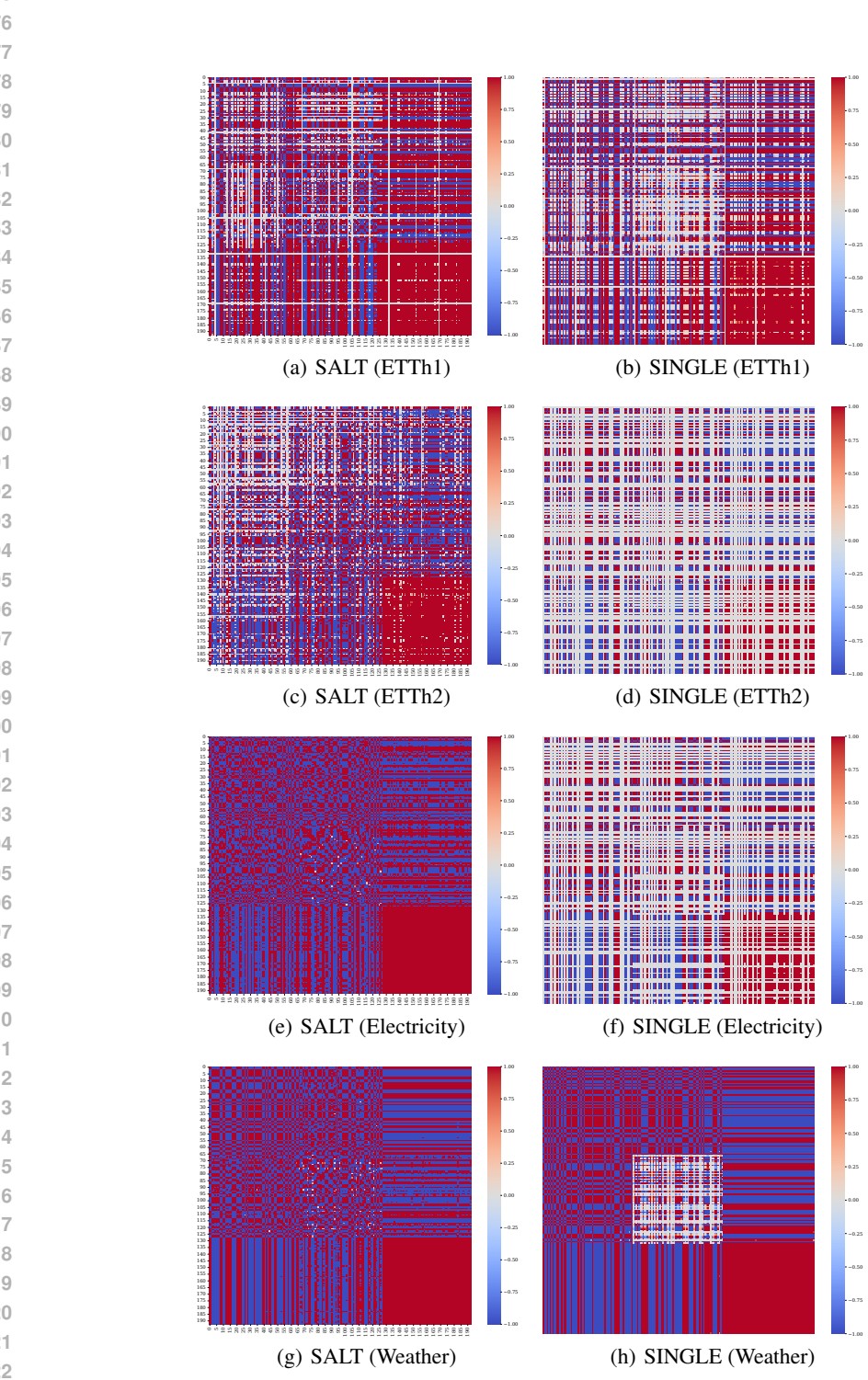

Figure 10: Visualization of the NTK Gram matrix on real-world datasets.

(a) SALT (ETTh1)  (b) SINGLE (ETTh1)

(c) SALT (ETTh2)  (d) SINGLE (ETTh2)

(e) SALT (Electricity)  (f) SINGLE (Electricity)

(g) SALT (Weather)  (h) SINGLE (Weather)

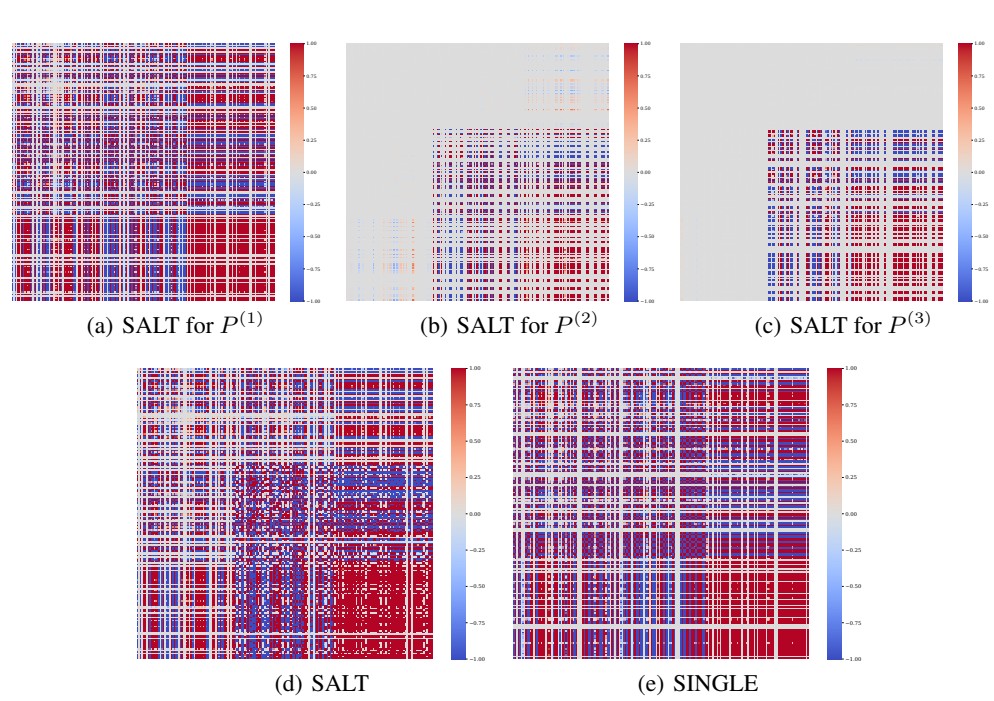

(a) SALT for $P^{(1)}$  (b) SALT for $P^{(2)}$  (c) SALT for $P^{(3)}$

(d) SALT  (e) SINGLE

Figure 11: Visualization of the NTK Gram matrix on synthetic datasets(constant noise).

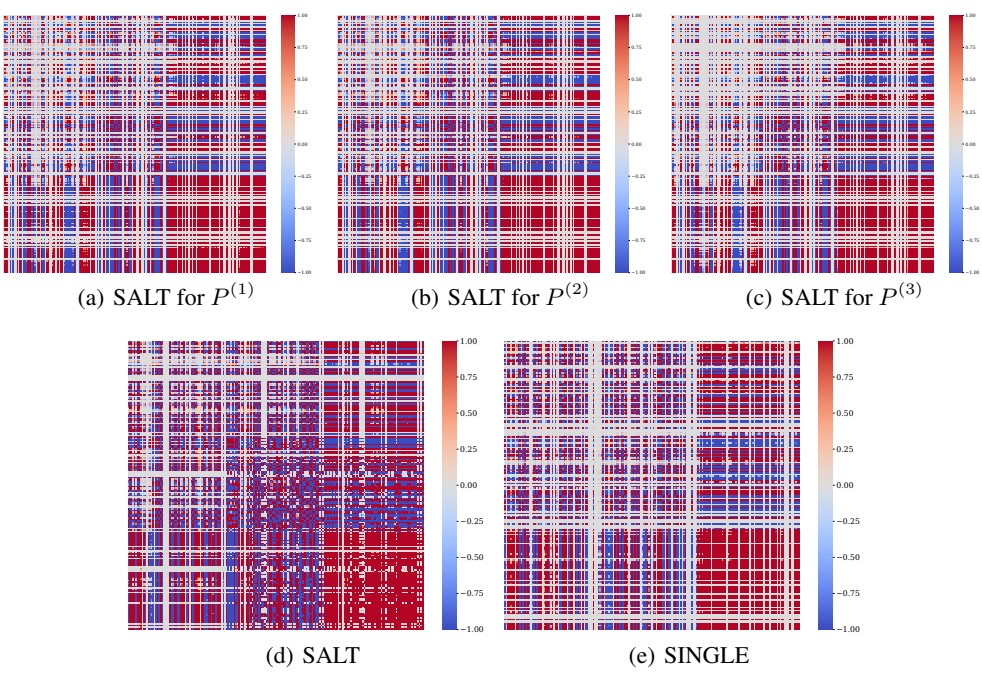

(a) SALT for $P^{(1)}$  (b) SALT for $P^{(2)}$  (c) SALT for $P^{(3)}$

(d) SALT  (e) SINGLE

Figure 12: Visualization of the NTK Gram matrix on synthetic datasets(Gaussian noise).

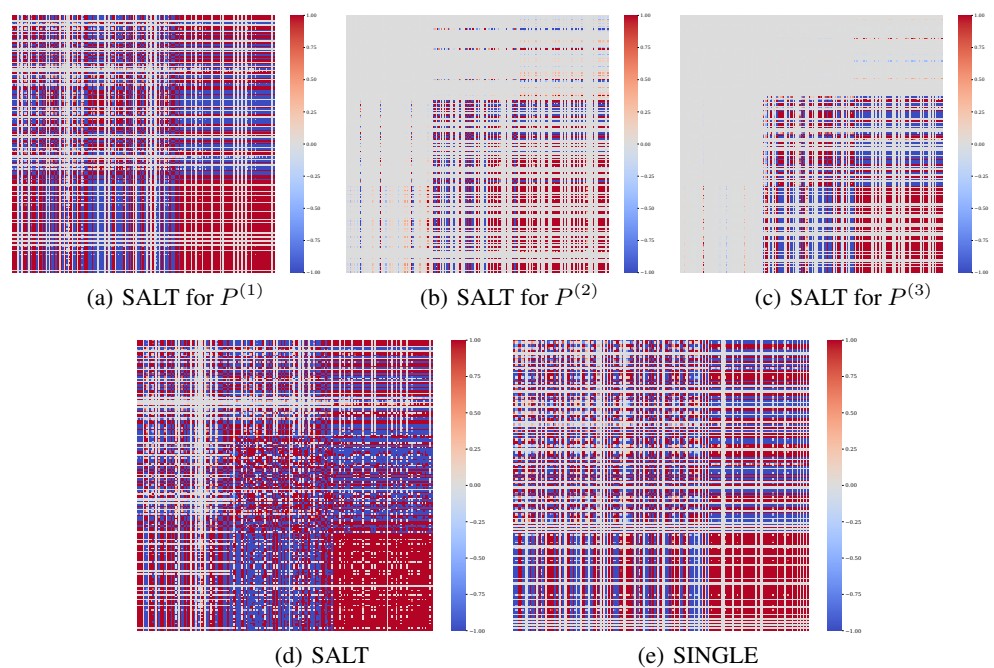

(a) SALT for $P^{(1)}$      (b) SALT for $P^{(2)}$      (c) SALT for $P^{(3)}$

(d) SALT          (e) SINGLE

Figure 13: Visualization of the NTK Gram matrix on synthetic datasets(Uniform noise).

## J  BROADER IMPACTS

Our proposed method, SALT, provides a flexible framework for enhancing time-series forecasting by structurally decomposing and separately training distinct signal components. This can benefit a wide range of applications, including economic forecasting (e.g., stock and cryptocurrency price prediction), infrastructure monitoring (e.g., traffic and energy demand), and public health surveillance (e.g., forecasting epidemic spread or hospital patient load).

These use cases may support critical decision-making in business, government, and healthcare. For instance, improved forecasting models can assist policymakers in responding to emerging trends or help clinicians anticipate patient outcomes. However, we caution that the use of our method in sensitive domains (e.g., medical or financial applications) must be accompanied by rigorous safeguards to ensure data privacy, transparency, and fairness.

We also acknowledge that forecasting models — if deployed irresponsibly — may exacerbate inequalities or be misused for speculative purposes. We encourage future work to incorporate ethical considerations and domain knowledge into both the modeling process and downstream deployment.

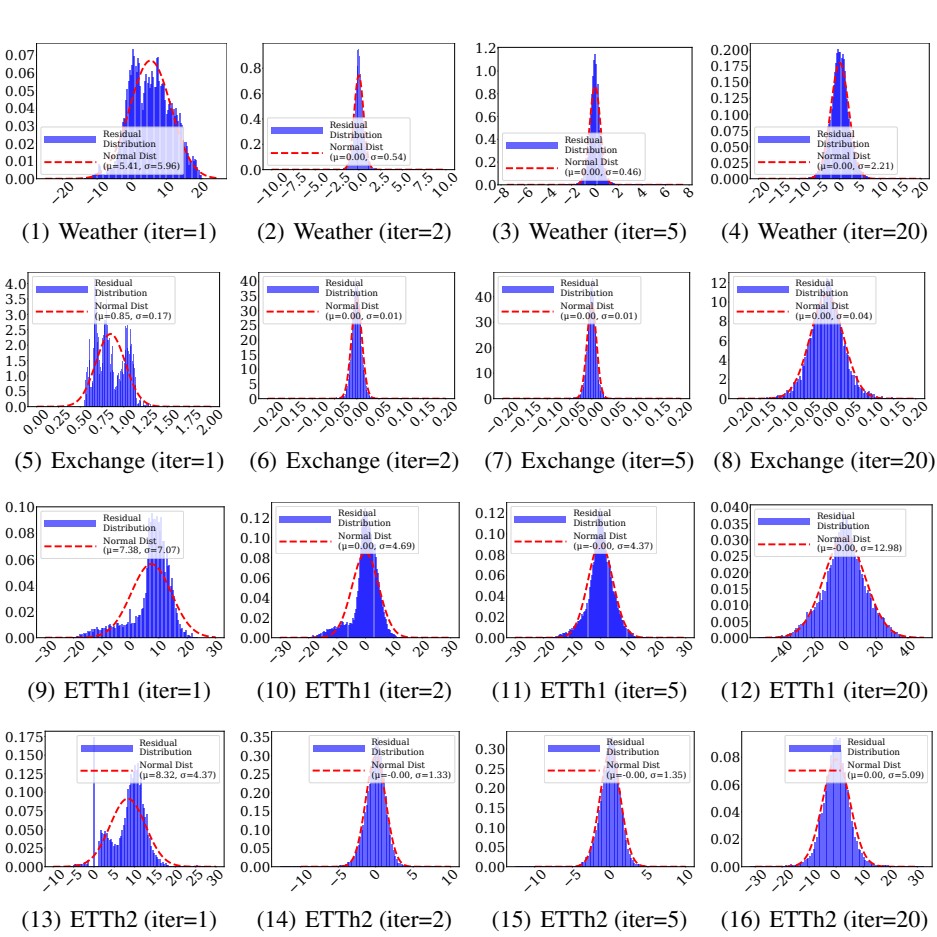

Figure 14: Visualization of the residual distributions over successive iterations for the 9 datasets. As the iteration process progresses, the residuals (blue bars) increasingly resemble a normal distribution (red dashed line), indicating that STL decomposition method iteratively refines the residuals towards a more Gaussian-like distribution.

Table 16: Results with 3 different random seeds. $h$ refers to the prediction horizon (or length).

| Datasets | | Weather | | Exchange | | ETTh1 | | ETTm1 | | ETTh2 | | ETTm2 | | Electricity | | Traffic | | | ILL | |
|---|---|---|---|---|---|---|---|---|---|---|---|---|---|---|---|---|---|---|---|---|
| | $h$ | MSE | MAE | MSE | MAE | MSE | MAE | MSE | MAE | MSE | MAE | MSE | MAE | MSE | MAE | MSE | MAE | $h$ | MSE | MAE |
| SALT N-BEATS | 96 | 0.164 ± 0.003 | 0.206 ± 0.003 | 0.089 ± 0.007 | 0.203 ± 0.003 | 0.315 ± 0.010 | 0.370 ± 0.007 | 0.248 ± 0.005 | 0.310 ± 0.002 | 0.287 ± 0.009 | 0.348 ± 0.017 | 0.179 ± 0.004 | 0.259 ± 0.004 | 0.144 ± 0.001 | 0.246 ± 0.002 | 0.395 ± 0.002 | 0.281 ± 0.001 | 24 | 1.834 ± 0.126 | 0.882 ± 0.009 |
| | 192 | 0.219 ± 0.008 | 0.259 ± 0.003 | 0.189 ± 0.005 | 0.313 ± 0.005 | 0.372 ± 0.004 | 0.404 ± 0.007 | 0.292 ± 0.004 | 0.342 ± 0.004 | 0.358 ± 0.001 | 0.389 ± 0.002 | 0.245 ± 0.008 | 0.306 ± 0.002 | 0.177 ± 0.002 | 0.284 ± 0.004 | 0.406 ± 0.002 | 0.291 ± 0.002 | 36 | 2.016 ± 0.093 | 0.958 ± 0.066 |
| | 336 | 0.267 ± 0.009 | 0.299 ± 0.004 | 0.393 ± 0.006 | 0.468 ± 0.012 | 0.401 ± 0.003 | 0.426 ± 0.007 | 0.318 ± 0.008 | 0.363 ± 0.005 | 0.357 ± 0.004 | 0.412 ± 0.006 | 0.306 ± 0.004 | 0.348 ± 0.008 | 0.196 ± 0.004 | 0.304 ± 0.003 | 0.438 ± 0.009 | 0.311 ± 0.006 | 48 | 2.025 ± 0.022 | 0.993 ± 0.091 |
| | 720 | 0.337 ± 0.003 | 0.345 ± 0.004 | 1.100 ± 0.008 | 0.800 ± 0.004 | 0.427 ± 0.012 | 0.457 ± 0.012 | 0.383 ± 0.008 | 0.397 ± 0.005 | 0.416 ± 0.000 | 0.432 ± 0.007 | 0.404 ± 0.007 | 0.411 ± 0.012 | 0.267 ± 0.003 | 0.359 ± 0.003 | 0.584 ± 0.004 | 0.385 ± 0.005 | 60 | 2.136 ± 0.016 | 0.999 ± 0.076 |
| SALT N-HiTS | 96 | 0.136 ± 0.001 | 0.177 ± 0.002 | 0.089 ± 0.002 | 0.194 ± 0.004 | 0.352 ± 0.002 | 0.388 ± 0.001 | 0.287 ± 0.002 | 0.332 ± 0.002 | 0.259 ± 0.001 | 0.341 ± 0.004 | 0.166 ± 0.003 | 0.244 ± 0.002 | 0.139 ± 0.001 | 0.223 ± 0.002 | 0.387 ± 0.001 | 0.276 ± 0.001 | 24 | 1.535 ± 0.006 | 0.786 ± 0.005 |
| | 192 | 0.209 ± 0.001 | 0.239 ± 0.000 | 0.188 ± 0.001 | 0.310 ± 0.000 | 0.385 ± 0.003 | 0.415 ± 0.002 | 0.329 ± 0.001 | 0.355 ± 0.002 | 0.337 ± 0.003 | 0.373 ± 0.002 | 0.232 ± 0.002 | 0.287 ± 0.001 | 0.159 ± 0.001 | 0.258 ± 0.001 | 0.403 ± 0.001 | 0.288 ± 0.001 | 36 | 1.966 ± 0.006 | 0.888 ± 0.001 |
| | 336 | 0.256 ± 0.001 | 0.301 ± 0.002 | 0.297 ± 0.001 | 0.388 ± 0.001 | 0.418 ± 0.002 | 0.433 ± 0.002 | 0.365 ± 0.012 | 0.399 ± 0.002 | 0.336 ± 0.000 | 0.367 ± 0.002 | 0.288 ± 0.001 | 0.315 ± 0.002 | 0.177 ± 0.001 | 0.288 ± 0.001 | 0.414 ± 0.001 | 0.303 ± 0.001 | 48 | 2.127 ± 0.002 | 0.912 ± 0.002 |
| | 720 | 0.337 ± 0.001 | 0.319 ± 0.001 | 0.686 ± 0.004 | 0.562 ± 0.001 | 0.423 ± 0.001 | 0.446 ± 0.002 | 0.418 ± 0.001 | 0.404 ± 0.001 | 0.403 ± 0.000 | 0.437 ± 0.002 | 0.386 ± 0.001 | 0.403 ± 0.002 | 0.214 ± 0.001 | 0.330 ± 0.006 | 0.506 ± 0.001 | 0.344 ± 0.000 | 60 | 2.022 ± 0.000 | 0.958 ± 0.001 |
| SALT DLinear | 96 | 0.147 ± 0.005 | 0.207 ± 0.001 | 0.076 ± 0.011 | 0.183 ± 0.010 | 0.241 ± 0.009 | 0.336 ± 0.002 | 0.246 ± 0.004 | 0.315 ± 0.003 | 0.269 ± 0.005 | 0.335 ± 0.003 | 0.150 ± 0.008 | 0.245 ± 0.006 | 0.127 ± 0.005 | 0.229 ± 0.002 | 0.340 ± 0.001 | 0.279 ± 0.005 | 24 | 1.863 ± 0.079 | 0.939 ± 0.021 |
| | 192 | 0.200 ± 0.002 | 0.264 ± 0.005 | 0.155 ± 0.005 | 0.284 ± 0.008 | 0.315 ± 0.002 | 0.377 ± 0.004 | 0.283 ± 0.002 | 0.338 ± 0.002 | 0.343 ± 0.002 | 0.394 ± 0.004 | 0.218 ± 0.006 | 0.296 ± 0.008 | 0.142 ± 0.000 | 0.248 ± 0.002 | 0.398 ± 0.002 | 0.294 ± 0.007 | 36 | 1.400 ± 0.029 | 0.809 ± 0.012 |
| | 336 | 0.255 ± 0.003 | 0.313 ± 0.006 | 0.303 ± 0.001 | 0.409 ± 0.006 | 0.357 ± 0.005 | 0.419 ± 0.004 | 0.317 ± 0.002 | 0.360 ± 0.001 | 0.429 ± 0.003 | 0.448 ± 0.012 | 0.270 ± 0.002 | 0.337 ± 0.002 | 0.159 ± 0.001 | 0.267 ± 0.005 | 0.262 ± 0.003 | 0.386 ± 0.088 | 48 | 1.443 ± 0.060 | 0.831 ± 0.006 |
| | 720 | 0.317 ± 0.005 | 0.366 ± 0.004 | 0.626 ± 0.012 | 0.621 ± 0.016 | 0.385 ± 0.002 | 0.444 ± 0.009 | 0.405 ± 0.002 | 0.418 ± 0.001 | 0.605 ± 0.006 | 0.543 ± 0.011 | 0.394 ± 0.004 | 0.419 ± 0.002 | 0.196 ± 0.004 | 0.294 ± 0.004 | 0.303 ± 0.008 | 0.404 ± 0.088 | 60 | 1.562 ± 0.056 | 0.881 ± 0.005 |
| SALT PatchTST | 96 | 0.128 ± 0.001 | 0.183 ± 0.001 | 0.067 ± 0.001 | 0.177 ± 0.001 | 0.267 ± 0.005 | 0.343 ± 0.001 | 0.243 ± 0.006 | 0.315 ± 0.008 | 0.260 ± 0.002 | 0.323 ± 0.001 | 0.163 ± 0.001 | 0.252 ± 0.001 | 0.116 ± 0.001 | 0.218 ± 0.001 | 0.359 ± 0.001 | 0.244 ± 0.003 | 24 | 1.082 ± 0.022 | 0.707 ± 0.004 |
| | 192 | 0.179 ± 0.001 | 0.226 ± 0.001 | 0.165 ± 0.001 | 0.288 ± 0.001 | 0.303 ± 0.002 | 0.367 ± 0.003 | 0.284 ± 0.003 | 0.340 ± 0.001 | 0.338 ± 0.001 | 0.375 ± 0.001 | 0.225 ± 0.000 | 0.293 ± 0.003 | 0.143 ± 0.000 | 0.235 ± 0.002 | 0.378 ± 0.002 | 0.253 ± 0.001 | 36 | 1.123 ± 0.005 | 0.730 ± 0.007 |
| | 336 | 0.235 ± 0.000 | 0.274 ± 0.001 | 0.327 ± 0.003 | 0.435 ± 0.004 | 0.335 ± 0.001 | 0.401 ± 0.002 | 0.309 ± 0.001 | 0.359 ± 0.001 | 0.327 ± 0.001 | 0.378 ± 0.002 | 0.272 ± 0.001 | 0.327 ± 0.001 | 0.158 ± 0.000 | 0.249 ± 0.001 | 0.394 ± 0.004 | 0.262 ± 0.001 | 48 | 1.294 ± 0.009 | 0.765 ± 0.010 |
| | 720 | 0.302 ± 0.001 | 0.326 ± 0.002 | 0.875 ± 0.003 | 0.695 ± 0.002 | 0.348 ± 0.001 | 0.416 ± 0.003 | 0.355 ± 0.001 | 0.389 ± 0.001 | 0.378 ± 0.001 | 0.418 ± 0.001 | 0.363 ± 0.002 | 0.384 ± 0.001 | 0.193 ± 0.001 | 0.287 ± 0.001 | 0.436 ± 0.005 | 0.284 ± 0.004 | 60 | 1.251 ± 0.037 | 0.764 ± 0.011 |
| SALT iTransformer | 96 | 0.140 ± 0.001 | 0.188 ± 0.002 | 0.075 ± 0.003 | 0.185 ± 0.002 | 0.295 ± 0.001 | 0.361 ± 0.005 | 0.268 ± 0.005 | 0.344 ± 0.002 | 0.293 ± 0.001 | 0.336 ± 0.001 | 0.181 ± 0.004 | 0.266 ± 0.002 | 0.142 ± 0.003 | 0.239 ± 0.001 | 0.317 ± 0.006 | 0.262 ± 0.007 | 24 | 1.794 ± 0.001 | 0.922 ± 0.004 |
| | 192 | 0.195 ± 0.001 | 0.235 ± 0.004 | 0.169 ± 0.000 | 0.291 ± 0.007 | 0.332 ± 0.006 | 0.385 ± 0.009 | 0.326 ± 0.005 | 0.371 ± 0.001 | 0.374 ± 0.028 | 0.398 ± 0.001 | 0.247 ± 0.005 | 0.307 ± 0.000 | 0.160 ± 0.003 | 0.258 ± 0.002 | 0.384 ± 0.007 | 0.272 ± 0.001 | 36 | 1.549 ± 0.001 | 0.866 ± 0.001 |
| | 336 | 0.249 ± 0.001 | 0.287 ± 0.003 | 0.329 ± 0.003 | 0.416 ± 0.005 | 0.360 ± 0.004 | 0.438 ± 0.012 | 0.345 ± 0.015 | 0.383 ± 0.004 | 0.404 ± 0.021 | 0.429 ± 0.006 | 0.311 ± 0.006 | 0.345 ± 0.003 | 0.172 ± 0.002 | 0.270 ± 0.001 | 0.413 ± 0.007 | 0.281 ± 0.001 | 48 | 1.569 ± 0.003 | 0.874 ± 0.012 |
| | 720 | 0.334 ± 0.005 | 0.341 ± 0.001 | 0.733 ± 0.008 | 0.642 ± 0.007 | 0.394 ± 0.007 | 0.451 ± 0.008 | 0.393 ± 0.008 | 0.423 ± 0.001 | 0.429 ± 0.002 | 0.443 ± 0.006 | 0.412 ± 0.009 | 0.407 ± 0.002 | 0.204 ± 0.002 | 0.306 ± 0.001 | 0.462 ± 0.003 | 0.310 ± 0.006 | 60 | 1.605 ± 0.002 | 0.877 ± 0.004 |
| SALT TimeMixer | 96 | 0.147 ± 0.002 | 0.196 ± 0.003 | 0.089 ± 0.001 | 0.211 ± 0.001 | 0.267 ± 0.007 | 0.343 ± 0.002 | 0.243 ± 0.010 | 0.327 ± 0.003 | 0.283 ± 0.005 | 0.345 ± 0.005 | 0.154 ± 0.003 | 0.248 ± 0.002 | 0.114 ± 0.002 | 0.217 ± 0.005 | 0.338 ± 0.003 | 0.283 ± 0.004 | 24 | 1.331 ± 0.014 | 0.772 ± 0.004 |
| | 192 | 0.205 ± 0.003 | 0.245 ± 0.002 | 0.173 ± 0.003 | 0.295 ± 0.002 | 0.332 ± 0.002 | 0.386 ± 0.003 | 0.278 ± 0.015 | 0.349 ± 0.001 | 0.348 ± 0.001 | 0.386 ± 0.003 | 0.237 ± 0.002 | 0.295 ± 0.005 | 0.154 ± 0.002 | 0.251 ± 0.005 | 0.396 ± 0.010 | 0.303 ± 0.012 | 36 | 1.078 ± 0.010 | 0.694 ± 0.009 |
| | 336 | 0.244 ± 0.001 | 0.279 ± 0.001 | 0.347 ± 0.002 | 0.424 ± 0.002 | 0.336 ± 0.003 | 0.396 ± 0.002 | 0.318 ± 0.001 | 0.377 ± 0.006 | 0.379 ± 0.001 | 0.411 ± 0.003 | 0.297 ± 0.001 | 0.342 ± 0.004 | 0.169 ± 0.001 | 0.267 ± 0.002 | 0.436 ± 0.011 | 0.298 ± 0.006 | 48 | 1.171 ± 0.018 | 0.760 ± 0.011 |
| | 720 | 0.336 ± 0.003 | 0.340 ± 0.001 | 0.952 ± 0.012 | 0.727 ± 0.003 | 0.385 ± 0.001 | 0.437 ± 0.002 | 0.374 ± 0.002 | 0.407 ± 0.003 | 0.413 ± 0.003 | 0.432 ± 0.004 | 0.385 ± 0.003 | 0.395 ± 0.002 | 0.216 ± 0.008 | 0.306 ± 0.002 | 0.439 ± 0.003 | 0.311 ± 0.003 | 60 | 1.458 ± 0.057 | 0.817 ± 0.019 |

Table 17: N-BEATS (SALT) Hyperparameters used for each dataset

| Datasets | Weather | | | | Exchange | | | | ETTh1 | | | |
|---|---|---|---|---|---|---|---|---|---|---|---|---|
| $h$ | 96 | 192 | 336 | 720 | 96 | 192 | 336 | 720 | 96 | 192 | 336 | 720 |
| $I$ | 336 | 336 | 336 | 336 | 192 | 192 | 192 | 192 | 192 | 192 | 192 | 192 |
| $k$ | 25 | 7 | 7 | 7 | 15 | 7 | 15 | 15 | 15 | 7 | 7 | 7 |
| $p$ | 12 | 12 | 12 | 12 | 12 | 12 | 12 | 12 | 12 | 12 | 12 | 12 |
| $\lambda$ | 0.00025 | 0.001 | 0.00025 | 0.00025 | 0.001 | 0.001 | 0.01 | 0.001 | 0.001 | 0.001 | 0.001 | 0.001 |
| Decomp. method | STL | STL | STL | STL | STL | STR | STR | STR | STL | STL | STL | STL |
| Datasets | ETTm1 | | | | ETTh2 | | | | ETTm2 | | | |
| $h$ | 96 | 192 | 336 | 720 | 96 | 192 | 336 | 720 | 96 | 192 | 336 | 720 |
| $I$ | 192 | 192 | 192 | 192 | 192 | 192 | 192 | 192 | 192 | 192 | 192 | 192 |
| $k$ | 7 | 7 | 24 | 24 | 15 | 13 | 15 | 15 | 7 | 7 | 47 | 47 |
| $p$ | 12 | 12 | 12 | 12 | 12 | 12 | 12 | 12 | 12 | 12 | 12 | 12 |
| $\lambda$ | 0.0001 | 0.001 | 0.001 | 0.001 | 0.001 | 0.01 | 0.005 | 0.01 | 0.001 | 0.001 | 0.001 | 0.001 |
| Decomp. method | STL | STL | STL | STL | STL | STL | STL | STL | STR | STR | STR | STR |
| Datasets | Electricity | | | | Traffic | | | | ILL | | | |
| $h$ | 96 | 192 | 336 | 720 | 96 | 192 | 336 | 720 | 24 | 36 | 48 | 60 |
| I | 336 | 336 | 336 | 336 | 336 | 336 | 336 | 336 | 104 | 104 | 104 | 104 |
| $k$ | 7 | 7 | 7 | 7 | 7 | 7 | 7 | 7 | 15 | 13 | 15 | 7 |
| $p$ | 12 | 12 | 12 | 12 | 12 | 12 | 12 | 12 | 12 | 12 | 12 | 12 |
| $\lambda$ | 0.01 | 0.01 | 0.01 | 0.01 | 0.01 | 0.01 | 0.01 | 0.01 | 0.001 | 0.001 | 0.001 | 0.001 |
| Decomp. method | STL | STL | STL | STL | STL | STL | STL | STL | STL | STL | STL | STL |

Table 18: N-HiTS (SALT) Hyperparameters used for each dataset

| Datasets | Weather | | | | Exchange | | | | ETTh1 | | | |
|---|---|---|---|---|---|---|---|---|---|---|---|---|
| $h$ | 96 | 192 | 336 | 720 | 96 | 192 | 336 | 720 | 96 | 192 | 336 | 720 |
| $I$ | 96 | 96 | 192 | 96 | 336 | 96 | 96 | 96 | 336 | 336 | 336 | 336 |
| $k$ | 14 | 14 | 14 | 14 | 7 | 12 | 12 | 12 | 9 | 9 | 43 | 47 |
| $p$ | 12 | 12 | 12 | 12 | 12 | 8 | 8 | 8 | 12 | 12 | 12 | 12 |
| $\lambda$ | 0.01 | 0.01 | 0.01 | 0.01 | 0.01 | 0.01 | 0.01 | 0.01 | 0.001 | 0.001 | 0.005 | 0.005 |
| Decomp. method | STR | STR | STR | STR | STL | STR | STR | STR | STL | STL | STL | STL |
| Datasets | ETTm1 | | | | ETTh2 | | | | ETTm2 | | | |
| $h$ | 96 | 192 | 336 | 720 | 96 | 192 | 336 | 720 | 96 | 192 | 336 | 720 |
| $I$ | 336 | 336 | 336 | 336 | 336 | 336 | 336 | 336 | 336 | 336 | 336 | 336 |
| $k$ | 24 | 24 | 24 | 24 | 48 | 9 | 23 | 9 | 47 | 47 | 47 | 47 |
| $p$ | 12 | 12 | 12 | 12 | 12 | 12 | 12 | 12 | 12 | 12 | 12 | 12 |
| $\lambda$ | 0.01 | 0.01 | 0.001 | 0.001 | 0.01 | 0.005 | 0.001 | 0.01 | 0.000125 | 0.001 | 0.001 | 0.001 |
| Decomp. method | STR | STR | STR | STR | STR | STL | STL | STL | STR | STR | STR | STR |
| Datasets | Electricity | | | | Traffic | | | | ILL | | | |
| $h$ | 96 | 192 | 336 | 720 | 96 | 192 | 336 | 720 | 24 | 36 | 48 | 60 |
| I | 336 | 336 | 336 | 336 | 336 | 336 | 336 | 336 | 144 | 144 | 104 | 104 |
| $k$ | 7 | 7 | 7 | 7 | 7 | 7 | 7 | 7 | 43 | 23 | 9 | 23 |
| $p$ | 12 | 12 | 12 | 12 | 12 | 12 | 12 | 12 | 24 | 12 | 12 | 12 |
| $\lambda$ | 0.01 | 0.01 | 0.01 | 0.01 | 0.01 | 0.01 | 0.01 | 0.01 | 0.005 | 0.01 | 0.001 | 0.01 |
| Decomp. method | STL | STL | STL | STL | STL | STL | STL | STL | STL | STL | STL | STL |

Table 19: DLinear (SALT) Hyperparameters used for each dataset.

| Datasets | Weather | | | | Exchange | | | | ETTh1 | | | |
|---|---|---|---|---|---|---|---|---|---|---|---|---|
| $h$ | 96 | 192 | 336 | 720 | 96 | 192 | 336 | 720 | 96 | 192 | 336 | 720 |
| $I$ | 336 | 336 | 336 | 480 | 336 | 336 | 336 | 336 | 336 | 336 | 192 | 336 |
| $k$ | 19 | 48 | 54 | 60 | 60 | 60 | 60 | 60 | 13 | 25 | 48 | 29 |
| $p$ | 12 | 6 | 6 | 36 | 7 | 7 | 7 | 7 | 12 | 12 | 24 | 12 |
| $\lambda$ | 0.001 | 0.001 | 0.001 | 0.00125 | 0.005 | 0.005 | 0.005 | 0.005 | 0.005 | 0.000125 | 0.000125 | 0.000125 |
| Decomp. method | STR | STR | STR | STR | STR | STR | STR | STR | STL | STL | STR | STL |
| Datasets | ETTm1 | | | | ETTh2 | | | | ETTm2 | | | |
| $h$ | 96 | 192 | 336 | 720 | 96 | 192 | 336 | 720 | 96 | 192 | 336 | 720 |
| I | 336 | 336 | 336 | 336 | 336 | 336 | 336 | 336 | 336 | 336 | 336 | 336 |
| $k$ | 24 | 12 | 12 | 12 | 42 | 36 | 42 | 42 | 48 | 48 | 48 | 48 |
| $p$ | 48 | 24 | 24 | 24 | 12 | 24 | 12 | 12 | 48 | 48 | 48 | 48 |
| $\lambda$ | 0.0025 | 0.00025 | 0.00025 | 0.0025 | 0.005 | 0.0025 | 0.005 | 0.005 | 0.0025 | 0.0025 | 0.0025 | 0.0025 |
| Decomp. method | STR | STR | STR | STR | STL | STR | STL | STL | STR | STR | STR | STR |
| Datasets | Electricity | | | | Traffic | | | | ILL | | | |
| $h$ | 96 | 192 | 336 | 720 | 96 | 192 | 336 | 720 | 24 | 36 | 48 | 60 |
| I | 336 | 336 | 336 | 336 | 336 | 720 | 336 | 336 | 104 | 104 | 104 | 104 |
| $k$ | 25 | 25 | 49 | 49 | 7 | 3 | 7 | 5 | 7 | 7 | 7 | 7 |
| $p$ | 12 | 12 | 12 | 12 | 12 | 12 | 12 | 12 | 12 | 12 | 12 | 12 |
| $\lambda$ | 0.0025 | 0.001 | 0.005 | 0.001 | 0.001 | 0.00025 | 0.001 | 0.025 | 0.005 | 0.005 | 0.005 | 0.005 |
| Decomp. method | STL | STL | STL | STL | STL | STR | STL | STL | STL | STL | STL | STL |

Table 20: PatchTST (SALT) Hyperparameters used for each dataset

| Datasets | Weather | | | | Exchange | | | | ETTh1 | | | |
|---|---|---|---|---|---|---|---|---|---|---|---|---|
| $h$ | 96 | 192 | 336 | 720 | 96 | 192 | 336 | 720 | 96 | 192 | 336 | 720 |
| $I$ | 336 | 336 | 336 | 720 | 336 | 336 | 336 | 336 | 336 | 336 | 336 | 336 |
| $k$ | 35 | 42 | 42 | 42 | 36 | 30 | 6 | 12 | 25 | 25 | 25 | 24 |
| $p$ | 42 | 14 | 35 | 28 | 24 | 6 | 7 | 6 | 12 | 12 | 12 | 24 |
| $\lambda$ | 0.0001 | 0.00001 | 0.0005 | 0.0005 | 0.00001 | 0.00025 | 0.000025 | 0.00025 | 0.000025 | 0.000025 | 0.000025 | 0.00001 |
| Decomp. method | STR | STR | STR | STR | STR | STR | STR | STR | STL | STL | STR | STL |
| Datasets | ETTm1 | | | | ETTh2 | | | | ETTm2 | | | |
| $h$ | 96 | 192 | 336 | 720 | 96 | 192 | 336 | 720 | 96 | 192 | 336 | 720 |
| I | 336 | 336 | 336 | 336 | 336 | 336 | 336 | 336 | 336 | 336 | 336 | 336 |
| $k$ | 7 | 7 | 7 | 7 | 42 | 9 | 23 | 19 | 11 | 11 | 11 | 11 |
| $p$ | 12 | 12 | 12 | 12 | 24 | 12 | 12 | 12 | 12 | 12 | 12 | 12 |
| $\lambda$ | 0.001 | 0.001 | 0.001 | 0.001 | 0.000125 | 0.0001 | 0.00025 | 0.001 | 0.0025 | 0.0025 | 0.0025 | 0.0025 |
| Decomp. method | STR | STR | STR | STR | STR | STL | STL | STL | STR | STR | STR | STR |
| Datasets | Electricity | | | | Traffic | | | | ILL | | | |
| $h$ | 96 | 192 | 336 | 720 | 96 | 192 | 336 | 720 | 24 | 36 | 48 | 60 |
| I | 336 | 336 | 336 | 336 | 336 | 336 | 336 | 336 | 104 | 104 | 104 | 104 |
| $k$ | 13 | 13 | 13 | 13 | 7 | 7 | 7 | 7 | 9 | 9 | 7 | 29 |
| $p$ | 12 | 12 | 12 | 12 | 12 | 12 | 12 | 12 | 12 | 12 | 12 | 12 |
| $\lambda$ | 0.0025 | 0.0025 | 0.0025 | 0.001 | 0.001 | 0.001 | 0.001 | 0.005 | 0.025 | 0.025 | 0.025 | 0.025 |
| Decomp. method | STL | STL | STL | STL | STL | STL | STL | STL | STL | STL | STL | STL |

Table 21: iTransformer (SALT) Hyperparameters used for each dataset

| Datasets | Weather | | | | Exchange | | | | ETTh1 | | | |
|---|---|---|---|---|---|---|---|---|---|---|---|---|
| $h$ | 96 | 192 | 336 | 720 | 96 | 192 | 336 | 720 | 96 | 192 | 336 | 720 |
| $I$ | 192 | 192 | 192 | 336 | 336 | 336 | 336 | 336 | 336 | 336 | 192 | 336 |
| $k$ | 48 | 48 | 48 | 48 | 60 | 60 | 60 | 60 | 9 | 23 | 24 | 24 |
| $p$ | 6 | 6 | 6 | 6 | 7 | 7 | 28 | 7 | 12 | 12 | 24 | 24 |
| $\lambda$ | 0.0005 | 0.005 | 0.005 | 0.005 | 0.000125 | 0.00025 | 0.001 | 0.001 | 0.000025 | 0.000125 | 0.000125 | 0.000125 |
| Decomp. method | STR | STR | STR | STR | STR | STR | STR | STR | STL | STL | STR | STR |

| Datasets | ETTm1 | | | | ETTh2 | | | | ETTm2 | | | |
|---|---|---|---|---|---|---|---|---|---|---|---|---|
| $h$ | 96 | 192 | 336 | 720 | 96 | 192 | 336 | 720 | 96 | 192 | 336 | 720 |
| $I$ | 336 | 336 | 336 | 336 | 336 | 336 | 336 | 336 | 336 | 336 | 336 | 336 |
| $k$ | 30 | 30 | 30 | 30 | 42 | 42 | 42 | 42 | 24 | 24 | 24 | 24 |
| $p$ | 24 | 24 | 24 | 24 | 24 | 24 | 24 | 24 | 24 | 24 | 24 | 24 |
| $\lambda$ | 0.0025 | 0.001 | 0.0025 | 0.001 | 0.000025 | 0.000025 | 0.000025 | 0.001 | 0.001 | 0.001 | 0.001 | 0.001 |
| Decomp. method | STR | STR | STR | STR | STR | STR | STR | STR | STR | STR | STR | STR |

| Datasets | Electricity | | | | Traffic | | | | ILL | | | |
|---|---|---|---|---|---|---|---|---|---|---|---|---|
| $h$ | 96 | 192 | 336 | 720 | 96 | 192 | 336 | 720 | 24 | 36 | 48 | 60 |
| I | 336 | 336 | 336 | 336 | 336 | 336 | 336 | 336 | 144 | 104 | 104 | 104 |
| $k$ | 7 | 7 | 7 | 7 | 9 | 9 | 9 | 9 | 7 | 9 | 7 | 7 |
| $p$ | 24 | 24 | 24 | 24 | 12 | 12 | 12 | 12 | 12 | 12 | 12 | 12 |
| $\lambda$ | 0.0001 | 0.0001 | 0.0001 | 0.0001 | 0.005 | 0.005 | 0.005 | 0.005 | 0.005 | 0.000025 | 0.000025 | 0.000025 |
| Decomp. method | STR | STR | STR | STR | STL | STL | STL | STL | STL | STL | STL | STL |

Table 22: TimeMixer (SALT) Hyperparameters used for each dataset

| Datasets | Weather | | | | Exchange | | | | ETTh1 | | | |
|---|---|---|---|---|---|---|---|---|---|---|---|---|
| $h$ | 96 | 192 | 336 | 720 | 96 | 192 | 336 | 720 | 96 | 192 | 336 | 720 |
| $I$ | 96 | 96 | 192 | 96 | 336 | 96 | 96 | 96 | 336 | 336 | 336 | 336 |
| $k$ | 14 | 14 | 14 | 14 | 7 | 12 | 12 | 12 | 9 | 9 | 43 | 47 |
| $p$ | 12 | 12 | 12 | 12 | 12 | 8 | 8 | 8 | 12 | 12 | 12 | 12 |
| $\lambda$ | 0.01 | 0.01 | 0.01 | 0.01 | 0.01 | 0.01 | 0.01 | 0.01 | 0.001 | 0.001 | 0.005 | 0.005 |
| Decomp. method | STR | STR | STR | STR | STL | STR | STR | STR | STL | STL | STL | STL |

| Datasets | ETTm1 | | | | ETTh2 | | | | ETTm2 | | | |
|---|---|---|---|---|---|---|---|---|---|---|---|---|
| $h$ | 96 | 192 | 336 | 720 | 96 | 192 | 336 | 720 | 96 | 192 | 336 | 720 |
| $I$ | 336 | 336 | 336 | 336 | 336 | 336 | 336 | 336 | 336 | 336 | 336 | 336 |
| $k$ | 24 | 24 | 24 | 24 | 48 | 9 | 23 | 9 | 47 | 47 | 47 | 47 |
| $p$ | 12 | 12 | 12 | 12 | 12 | 12 | 12 | 12 | 12 | 12 | 12 | 12 |
| $\lambda$ | 0.01 | 0.01 | 0.001 | 0.001 | 0.01 | 0.005 | 0.001 | 0.01 | 0.000125 | 0.001 | 0.001 | 0.001 |
| Decomp. method | STR | STR | STR | STR | STR | STL | STL | STL | STR | STR | STR | STR |

| Datasets | Electricity | | | | Traffic | | | | ILL | | | |
|---|---|---|---|---|---|---|---|---|---|---|---|---|
| $h$ | 96 | 192 | 336 | 720 | 96 | 192 | 336 | 720 | 24 | 36 | 48 | 60 |
| I | 336 | 336 | 336 | 336 | 336 | 336 | 336 | 336 | 144 | 144 | 104 | 104 |
| $k$ | 7 | 7 | 7 | 7 | 7 | 7 | 7 | 7 | 43 | 23 | 9 | 23 |
| $p$ | 12 | 12 | 12 | 12 | 12 | 12 | 12 | 12 | 24 | 12 | 12 | 12 |
| $\lambda$ | 0.01 | 0.01 | 0.01 | 0.01 | 0.01 | 0.01 | 0.01 | 0.01 | 0.005 | 0.01 | 0.001 | 0.01 |
| Decomp. method | STL | STL | STL | STL | STL | STL | STL | STL | STL | STL | STL | STL |

Table 23: Computational time per 1 epoch

| Models | Weather | Exchange | ETTh1 | ETTm1 | ETTh2 | ETTm2 | Electricity | Traffic | ILL |
|---|---|---|---|---|---|---|---|---|---|
| N-BEATS | 31.18 | 8.711 | 5.398 | 69.35 | 10.15 | 63.74 | 34.89 | 33.52 | 0.634 |
| N-BEATS (SALT) | 27.70 | 7.070 | 4.822 | 63.74 | 11.42 | 69.35 | 33.45 | 31.15 | 0.559 |
| N-HiTS | 16.23 | 6.328 | 4.632 | 42.17 | 7.382 | 43.18 | 21.23 | 27.47 | 0.482 |
| N-HiTS (SALT) | 16.21 | 5.458 | 3.827 | 41.72 | 6.219 | 42.61 | 20.98 | 25.33 | 0.503 |
| DLinear | 7.824 | 3.699 | 1.898 | 6.371 | 2.011 | 4.324 | 9.964 | 13.78 | 1.186 |
| DLinear (SALT) | 10.70 | 3.458 | 3.339 | 4.383 | 3.518 | 8.492 | 14.37 | 12.40 | 0.956 |
| PatchTST | 79.57 | 5.297 | 21.31 | 10.81 | 22.95 | 4.512 | 208.6 | 312.8 | 3.574 |
| PatchTST (SALT) | 75.42 | 5.391 | 3.881 | 11.27 | 2.967 | 3.812 | 218.9 | 316.7 | 2.002 |
| iTransformer | 21.30 | 7.254 | 16.84 | 30.28 | 6.782 | 35.91 | 42.88 | 89.84 | 6.884 |
| iTransformer (SALT) | 21.45 | 6.687 | 14.49 | 27.39 | 5.294 | 36.84 | 35.56 | 117.9 | 7.217 |
| TimeMixer | 48.51 | 11.90 | 31.88 | 201.8 | 49.21 | 72.81 | 314.9 | 901.5 | 5.073 |
| TimeMixer (SALT) | 40.65 | 5.179 | 15.27 | 76.11 | 16.20 | 25.31 | 213.7 | 673.1 | 1.544 |

Table 24: Model usage (MB)

| Models | Weather | Exchange | ETTh1 | ETTm1 | ETTh2 | ETTm2 | Electricity | Traffic | ILL |
|---|---|---|---|---|---|---|---|---|---|
| N-BEATS | 48.11 | 47.23 | 30.17 | 30.17 | 30.17 | 30.17 | 208.8 | 383.4 | 20.84 |
| N-BEATS (SALT) | 48.69 | 47.23 | 30.17 | 30.17 | 30.17 | 30.17 | 208.8 | 383.4 | 20.84 |
| N-HiTS | 32.63 | 31.84 | 22.83 | 22.83 | 22.83 | 22.83 | 115.2 | 189.2 | 14.21 |
| N-HiTS (SALT) | 32.63 | 31.84 | 22.83 | 22.83 | 22.83 | 22.83 | 115.2 | 189.2 | 14.21 |
| DLinear | 21.87 | 19.11 | 23.35 | 19.09 | 23.35 | 23.35 | 67.61 | 148.8 | 17.89 |
| DLinear (SALT) | 21.87 | 19.11 | 20.90 | 21.00 | 23.35 | 23.35 | 67.61 | 116.3 | 17.89 |
| PatchTST | 4,102 | 398.5 | 571.5 | 84.57 | 370.0 | 452.8 | 1,543 | 1,628 | 170.1 |
| PatchTST (SALT) | 4,071 | 401.4 | 569.0 | 80.82 | 367.4 | 468.5 | 1,058 | 1,540 | 63.05 |
| iTransformer | 35.53 | 98.21 | 121.8 | 123.7 | 128.7 | 122.9 | 601.8 | 3,821 | 80.21 |
| iTransformer (SALT) | 253.2 | 163.2 | 198.1 | 159.7 | 158.8 | 159.7 | 866.6 | 5,336 | 186.8 |
| TimeMixer | 2,011 | 902.4 | 5,803 | 178.5 | 101.8 | 614.9 | 980.4 | 6,109 | 140.8 |
| TimeMixer (SALT) | 2,338 | 1,129 | 7,483 | 253.4 | 127.3 | 828.2 | 1,268 | 8,546 | 165.0 |

Table 25: Stationarity testing results on benchmarked datasets. An 'X' indicates that both ADF and KPSS tests agree the dataset is non-stationary, while △ denotes that at least one of the tests indicates non-stationarity. We refer to cases marked with △ as weakly non-stationary.

| | Weather | Exchange | ETTh1, ETTm1 | ETTh2, ETTm2 | Electricity | Traffic | ILL |
|---|---|---|---|---|---|---|---|
| Stationarity | X | X | X | △ | △ | △ | △ |

Table 26: Stationarity testing result on weather each variable using ADF and KPSS tests

| Variable | ADF | | | KPSS | | |
|---|---|---|---|---|---|---|
| | Statistic | p-value | Stationarity | Statistic | p-value | Stationarity |
| p (mbar) | -8.140 | 1.04e-12 | O | 0.9420 | 0.01 | X |
| T (degC) | -8.407 | 2.15e-13 | O | 8.1786 | 0.01 | X |
| Tpot (K) | -8.430 | 1.88e-13 | O | 8.1251 | 0.01 | X |
| Tdew (degC) | -6.505 | 1.14e-08 | O | 10.2832 | 0.01 | X |
| rh (%) | -17.053 | 8.04e-30 | O | 8.1633 | 0.01 | X |
| VPmax (mbar) | -9.398 | 6.31e-16 | O | 7.9154 | 0.01 | X |
| VPact (mbar) | -6.155 | 7.39e-08 | O | 9.3269 | 0.01 | X |
| VPdef (mbar) | -16.633 | 1.66e-29 | O | 5.0522 | 0.01 | X |
| sh (g/kg) | -6.177 | 6.58e-08 | O | 9.2933 | 0.01 | X |
| H2OC (mmol/mol) | -6.175 | 6.65e-08 | O | 9.3067 | 0.01 | X |
| rho (g/m$^3$) | -7.944 | 3.26e-12 | O | 7.5689 | 0.01 | X |
| wv (m/s) | -229.279 | 0.0 | O | 0.0207 | 0.1 | O |
| max. wv (m/s) | -22.467 | 0.0 | O | 3.2738 | 0.01 | X |
| wd (deg) | -18.273 | 2.32e-30 | O | 0.8335 | 0.01 | X |
| rain (mm) | -29.619 | 0.0 | O | 0.1776 | 0.1 | O |
| raining (s) | -21.262 | 0.0 | O | 0.1328 | 0.1 | O |
| SWDR (W/m$^2$) | -35.205 | 0.0 | O | 7.2054 | 0.01 | X |
| PAR (μmol/m$^2$/s) | -35.376 | 0.0 | O | 7.3854 | 0.01 | X |
| max. PAR (μmol/m$^2$/s) | -34.085 | 0.0 | O | 7.6197 | 0.01 | X |
| Tlog (degC) | -8.614 | 6.37e-14 | O | 8.3066 | 0.01 | X |
| OT | -25.113 | 0.0 | O | 0.2279 | 0.1 | O |

Table 27: Stationarity testing result on exchange rate each variable using ADF and KPSS tests

| Variable | ADF | | | KPSS | | |
|---|---|---|---|---|---|---|
| | Statistic | p-value | Stationarity | Statistic | p-value | Stationarity |
| 0 | -1.665 | 0.4492 | X | 5.2917 | 0.01 | X |
| 1 | -2.150 | 0.2250 | X | 1.2468 | 0.01 | X |
| 2 | -1.353 | 0.6048 | X | 5.3135 | 0.01 | X |
| 3 | -1.587 | 0.4903 | X | 10.5355 | 0.01 | X |
| 4 | -2.869 | 0.0491 | O | 2.4475 | 0.01 | X |
| 5 | -2.120 | 0.2365 | X | 3.9282 | 0.01 | X |
| 6 | -1.748 | 0.4067 | X | 7.9789 | 0.01 | X |
| OT | -1.728 | 0.4166 | X | 7.0661 | 0.01 | X |

Table 28: Stationarity testing result on ETTh1 each variable using ADF and KPSS tests

| Variable | ADF | | | KPSS | | |
|---|---|---|---|---|---|---|
| | Statistic | p-value | Stationarity | Statistic | p-value | Stationarity |
| HUFL | -8.5505 | 9.25e-14 | O | 6.4781 | 0.01 | X |
| HULL | -5.1691 | 1.02e-05 | O | 2.2266 | 0.01 | X |
| MUFL | -8.6212 | 6.10e-14 | O | 9.0633 | 0.01 | X |
| MULL | -4.9641 | 2.61e-05 | O | 1.7190 | 0.01 | X |
| LUFL | -5.7969 | 4.73e-07 | O | 2.0147 | 0.01 | X |
| LULL | -4.7727 | 6.13e-05 | O | 1.1559 | 0.01 | X |
| OT | -3.4880 | 0.0083 | O | 9.4621 | 0.01 | X |

Table 29: Stationarity testing result on ETTh2 each variable using ADF and KPSS tests

| Variable | ADF | | | KPSS | | |
|---|---|---|---|---|---|---|
| | Statistic | p-value | Stationarity | Statistic | p-value | Stationarity |
| HUFL | -6.5264 | 1.01e-08 | O | 8.8581 | 0.01 | X |
| HULL | -4.5542 | 0.0002 | O | 12.7572 | 0.01 | X |
| MUFL | -4.0446 | 0.0012 | O | 1.2424 | 0.01 | X |
| MULL | -4.4530 | 0.0002 | O | 8.1632 | 0.01 | X |
| LUFL | -2.4355 | 0.1320 | X | 16.5272 | 0.01 | X |
| LULL | -3.3408 | 0.0132 | O | 1.0814 | 0.01 | X |
| OT | -3.5971 | 0.0058 | O | 1.8443 | 0.01 | X |

Table 30: Stationarity testing result on electricity each variable using ADF and KPSS tests (showing statistics for a sample of 20 out of the total 321 variables).

| Variable | ADF | | | KPSS | | |
|---|---|---|---|---|---|---|
| | Statistic | p-value | Stationarity | Statistic | p-value | Stationarity |
| 0 | -6.8575 | 1.63e-09 | O | 1.0495 | 0.01 | X |
| 1 | -7.0165 | 6.71e-10 | O | 1.7130 | 0.01 | X |
| 2 | -5.0386 | 1.86e-05 | O | 4.6124 | 0.01 | X |
| 3 | -6.1206 | 8.87e-08 | O | 15.8351 | 0.01 | X |
| 4 | -4.9192 | 3.20e-05 | O | 1.3654 | 0.01 | X |
| 5 | -4.6755 | 9.35e-05 | O | 1.4239 | 0.01 | X |
| 6 | -6.6138 | 6.28e-09 | O | 0.6762 | 0.0157 | X |
| 7 | -12.2476 | 9.70e-23 | O | 1.2862 | 0.01 | X |
| 8 | -10.6022 | 6.13e-19 | O | 1.3742 | 0.01 | X |
| 9 | -12.0190 | 3.05e-22 | O | 1.2859 | 0.01 | X |
| 10 | -5.0968 | 1.42e-05 | O | 1.0898 | 0.01 | X |
| 11 | -22.8691 | 0.0 | O | 8.1358 | 0.01 | X |
| 12 | -15.0668 | 8.84e-28 | O | 0.6601 | 0.0172 | X |
| 13 | -7.5981 | 2.43e-11 | O | 0.7042 | 0.0132 | X |
| 14 | -6.5456 | 9.11e-09 | O | 0.8886 | 0.01 | X |
| 15 | -11.2439 | 1.78e-20 | O | 0.8630 | 0.01 | X |
| 16 | -8.4163 | 2.04e-13 | O | 6.2602 | 0.01 | X |
| 17 | -7.3335 | 1.11e-10 | O | 8.7768 | 0.01 | X |
| 18 | -8.7300 | 3.21e-14 | O | 2.9707 | 0.01 | X |
| 19 | -14.8180 | 1.98e-27 | O | 3.8465 | 0.01 | X |

Table 31: Stationarity testing result on traffic each variable using ADF and KPSS tests (showing statistics for a sample of 20 out of the total 883 variables).

| Variable | ADF | | | KPSS | | |
|---|---|---|---|---|---|---|
| | Statistic | p-value | Stationarity | Statistic | p-value | Stationarity |
| 0 | -15.1612 | 6.59e-28 | O | 1.1652 | 0.01 | X |
| 1 | -15.8957 | 8.44e-29 | O | 0.2681 | 0.1 | O |
| 2 | -15.0168 | 1.04e-27 | O | 0.9598 | 0.01 | X |
| 3 | -19.1828 | 0.00 | O | 0.6013 | 0.0225 | X |
| 4 | -16.9776 | 9.06e-30 | O | 0.1101 | 0.1 | O |
| 5 | -18.1970 | 2.41e-30 | O | 1.1706 | 0.01 | X |
| 6 | -20.0680 | 0.00 | O | 1.8860 | 0.01 | X |
| 7 | -16.0007 | 6.52e-29 | O | 0.6163 | 0.0212 | X |
| 8 | -14.8895 | 1.57e-27 | O | 5.0434 | 0.01 | X |
| 9 | -15.2753 | 4.66e-28 | O | 1.1001 | 0.01 | X |
| 10 | -16.6232 | 1.69e-29 | O | 0.4720 | 0.0480 | X |
| 11 | -14.2461 | 1.51e-26 | O | 2.9375 | 0.01 | X |
| 12 | -16.9995 | 8.75e-30 | O | 1.6541 | 0.01 | X |
| 13 | -17.3764 | 5.10e-30 | O | 3.2661 | 0.01 | X |
| 14 | -17.3334 | 5.39e-30 | O | 2.8093 | 0.01 | X |
| 15 | -16.2227 | 3.88e-29 | O | 2.7122 | 0.01 | X |
| 16 | -10.2786 | 3.83e-18 | O | 3.0002 | 0.01 | X |
| 17 | -13.0770 | 1.90e-24 | O | 3.4275 | 0.01 | X |
| 18 | -15.6792 | 1.48e-28 | O | 2.6290 | 0.01 | X |
| 19 | -13.6018 | 1.95e-25 | O | 1.4351 | 0.01 | X |

Table 32: Stationarity testing result on national illness(ILL) each variable using ADF and KPSS tests

| Variable | ADF | | | KPSS | | |
|---|---|---|---|---|---|---|
| | Statistic | p-value | Stationarity | Statistic | p-value | Stationarity |
| % WEIGHTED | -7.846 | 5.75e-12 | O | 0.1857 | 0.1 | O |
| % UNWEIGHTED | -7.747 | 1.03e-11 | O | 0.3008 | 0.1 | O |
| AGE 0-4 | -6.507 | 1.12e-08 | O | 1.6179 | 0.01 | X |
| AGE 5-24 | -6.383 | 2.20e-08 | O | 1.2751 | 0.01 | X |
| ILITOTAL | -6.161 | 7.16e-08 | O | 1.7705 | 0.01 | X |
| # OF PROVIDERS | -1.713 | 0.4243 | X | 3.6161 | 0.01 | X |
| OT | -0.982 | 0.7598 | X | 4.0449 | 0.01 | X |

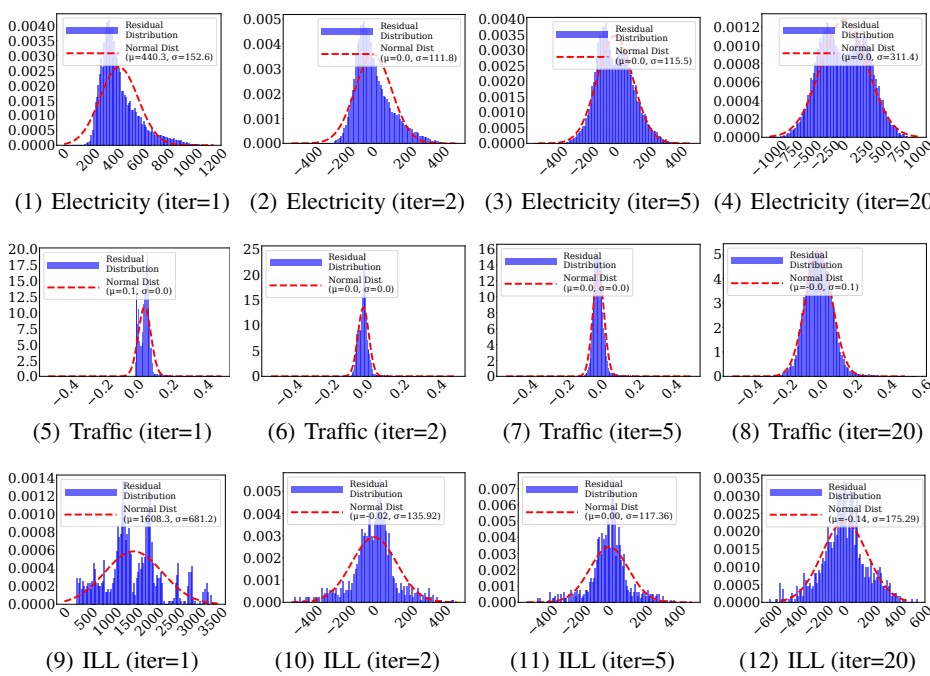

(1) Electricity (iter=1)    (2) Electricity (iter=2)    (3) Electricity (iter=5)    (4) Electricity (iter=20)

(5) Traffic (iter=1)    (6) Traffic (iter=2)    (7) Traffic (iter=5)    (8) Traffic (iter=20)

(9) ILL (iter=1)    (10) ILL (iter=2)    (11) ILL (iter=5)    (12) ILL (iter=20)

Figure 15: Visualization of the residual distributions over successive iterations for the 9 datasets. As the iteration process progresses, the residuals (blue bars) increasingly resemble a normal distribution (red dashed line), indicating that STL decomposition method iteratively refines the residuals towards a more Gaussian-like distribution.

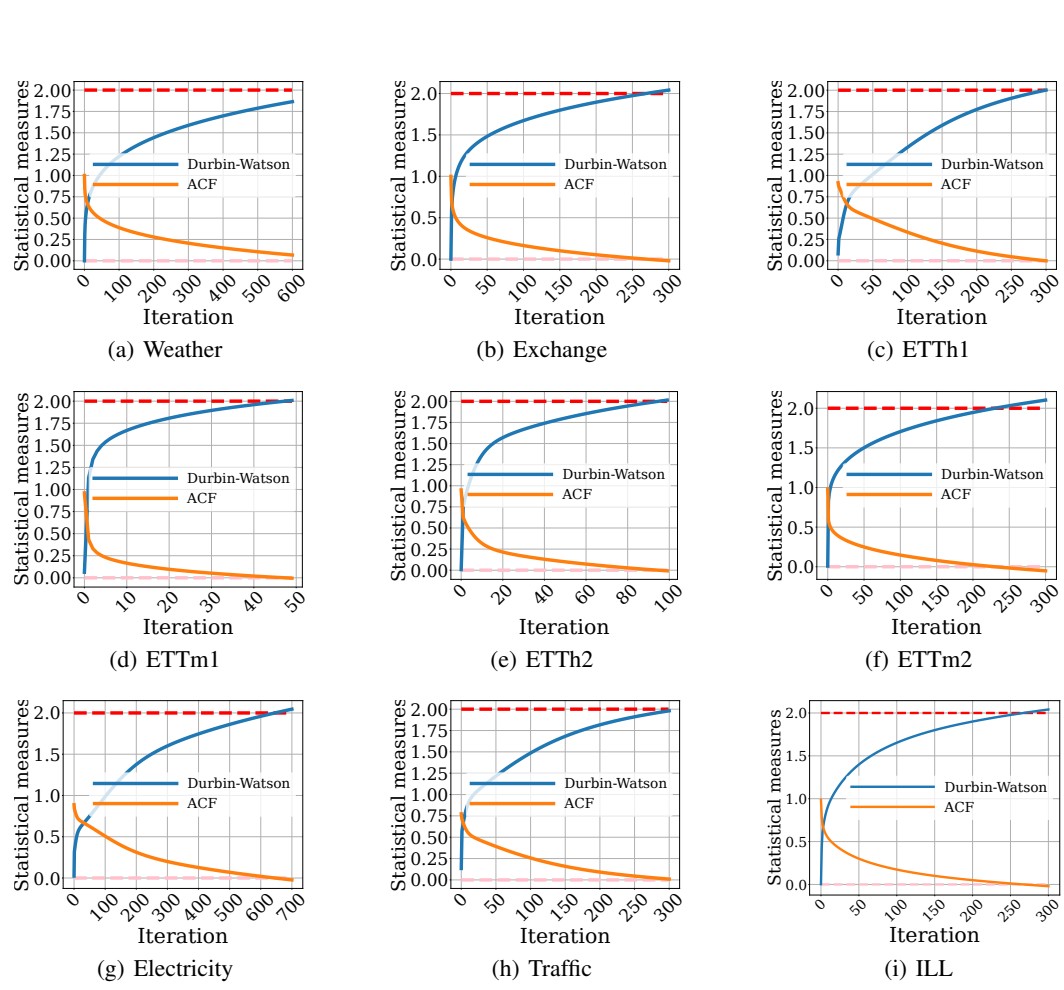

Figure 16: Empirical validation showing that as iterations increase, the residuals approach white noise, demonstrated by the Durbin-Watson and ACF statistics converging towards their expectation.

