# OpenReview forum: "SALT: Structure-Aligned Learning for Time-Series Forecasting"
_ICLR.cc/2026/Conference — Submitted to ICLR 2026_

### Official Review · Reviewer_1DLx · 2025-10-15

**Soundness:** 3
**Presentation:** 3
**Contribution:** 3
**Rating:** 6
**Confidence:** 5

**Summary:**

This paper introduces SALT (Structure-Aligned Learning for Time-Series Forecasting), a training paradigm that addresses gradient interference and spectral bias in time-series models. By combining Iterative Dominant Extraction (IDE) and Separable Training, SALT trains decomposed components independently and then aggregates them.

**Strengths:**

- Clear identification of two fundamental optimization challenges: gradient interference and spectral bias.

- Method is simple, general, and model-agnostic, applicable to diverse forecasting backbones.

- Provides solid theoretical justification that strengthens the contribution.

- Experimental evaluation is comprehensive: multiple datasets, architectures, and ablation studies demonstrate consistent gains.

- Improves both accuracy and interpretability by aligning optimization with structural decomposition.

**Weaknesses:**

- Could you analyze the performance of your method in the frequency domain as well? The current experiment are all conducted on the model that only includes the time domain, but many recent models transform data into the frequency domain (e.g., via Fourier transforms) for learning. It would be valuable to investigate whether similar issues also arise in the frequency space and how your approach might address them.

- In Table 24, why does your method incur such a large memory increase on the iTransformer model, while the overhead remains small for other backbones? Please clarify the cause of this discrepancy.

- The authors may consider discussing this work[1], which propose a hybrid loss framework combining the global and component losses, dynamically adjusting weights between these overall loss; GCformer[2], which identifies a gap or conflict between features extracted in the time domain and frequency domain, and then train them together to bridge this gap. It would be helpful to include a comparison in the related work section to explain the distinctions between these two methods and the current approach.

[1]Han, R., Feng, D., Du, H., Wang, H. (2025). A Hybrid Loss Framework for Decomposition-Based Time Series Forecasting Methods: Balancing Global and Component Errors. In: Wu, X., et al. Advances in Knowledge Discovery and Data Mining. PAKDD 2025. Lecture Notes in Computer Science(), vol 15873. Springer, Singapore. https://doi.org/10.1007/978-981-96-8183-9_15

[2] Yanjun Zhao, Ziqing` Ma, Tian Zhou, Mengni Ye, Liang Sun, and Yi Qian. 2023. GCformer: An Efficient Solution for Accurate and Scalable Long-Term Multivariate Time Series Forecasting. In Proceedings of the 32nd ACM International Conference on Information and Knowledge Management (CIKM '23). Association for Computing Machinery, New York, NY, USA, 3464–3473. https://doi.org/10.1145/3583780.3615136

**Questions:**

Same as weakness.

---

### Official Review · Reviewer_3iUt · 2025-10-28

**Soundness:** 2
**Presentation:** 2
**Contribution:** 1
**Rating:** 0
**Confidence:** 5

**Summary:**

SALT aims to addresse gradient interference and spectral bias in time-series forecasting by decomposing data into interpretable components with dedicated predictors. Theoretical results using structure-wise decoupling and NTK spectral analysis are provided to support this approach. Experiments on multiple datasets and models show improved accuracy and training stability.

**Strengths:**

The paper is clearly written and illustrated with explanatory figures.

**Weaknesses:**

1. The proof of Theorem 1 (Error Reduction via Structure-wise Decoupling) comes across as potentially misleading. In particular, the second point of the “the following two reasonable assumptions” (L674–L680) lacks practical justification. It is unclear how one can reasonably assume that separable training will consistently achieve lower empirical training loss than a single model, which seems unfounded. This argument essentially assumes that any multi-model prediction paradigm will “reasonably” perform better, which does not reflect the realities of neural network training. Such a claim oversimplifies the complex dynamics involved in optimization and generalization.

2. Similarly, the proof of Theorem 2 raises concerns. Prior to the NTK-based analysis, much of the derivation appears to involve manipulation of statistical quantities such as expectations. For instance, if $p_l$ is treated as a stochastic variable, it is not clear how it can be legitimately pulled out of the expectation in Eqs. 28–29. Assumptions should not be introduced merely to simplify derivations or produce desired results.

3. Finally, the benchmark datasets used are relatively old and small, which may allow the models to easily saturate the dynamics, limiting the generalizability of the results.

**Questions:**

Please see the [Weakness] section.

---

### Official Review · Reviewer_tWHh · 2025-11-01

**Soundness:** 3
**Presentation:** 3
**Contribution:** 2
**Rating:** 4
**Confidence:** 3

**Summary:**

The paper proposes SALT (Structure-Aligned Learning for Time-Series), which decomposes a time series into frequency-aligned components via Iterative Dominant Extraction and trains each component independently (Separable Training). The authors argue that single-loss training suffers from gradient interference and spectral bias, and claim SALT mitigates both through structure-wise decoupling.

**Strengths:**

- Identifies and formalizes two genuine optimization issues (gradient interference, spectral bias) often overlooked in forecasting.
- Consistent performance gains across diverse models suggest broad applicability.

**Weaknesses:**

- The central idea (training decomposed components separately) is conceptually close to existing frequency- or residual-based decompositions (e.g., FEDformer, N-HiTS, or frequency-domain ensembling). The incremental distinction between “Separable Training” and prior modular approaches is not convincingly articulated.
- Separable training multiplies model cost by the number of components; the paper does not analyze compute scaling or fairness versus single-model baselines.
- Theoretical results (Theorem 1–2) are mostly restatements of intuitive properties (removing cross-terms) and assume ideal independence between components..

**Questions:**

- How is this different in essence from training separate frequency-band or residual models already common in decomposition-based forecasting? Does SALT provide a new optimization mechanism beyond architectural modularization?
- What is the computational overhead of Separable Training (parameter count × N, training time)? Could similar gradient-decoupling effects be achieved with shared-parameter multi-head designs?
- Would partial parameter sharing (e.g., shared encoder, separate decoders) preserve the claimed theoretical benefits?
- Why does Figure 5 show minimal improvement on Weather (which is non-stationary), if spectral bias reduction is the main claim?

---

### Meta-Review · Area_Chair_z2vk · 2025-12-20

**Summary:**

This research investigates a structure decomposition method for seperated learning in time-series modeling. The reviewers acknowledge that the paper identifies two important and empirically observable issues in decomposed time-series forecasting, gradient interference and spectral bias, and proposes a clear training regime (SALT) that decouples component optimization via iterative dominant extraction and separable training. The empirical evaluation across multiple backbones and benchmarks is generally solid and demonstrates consistent performance gains over conventional single-objective training.

However, reviewers raised substantive concerns regarding the novelty and conceptual distinction of the proposed separable training strategy relative to prior decomposition-based or frequency-residual modeling approaches. Several reviewers questioned whether SALT introduces a fundamentally new optimization principle beyond existing modular or frequency-wise training paradigms, or whether it is primarily a reformulation with stronger empirical validation. In addition, there is significant concern about the theoretical rigor and assumptions underlying Theorems 1 and 2. Reviewers noted that key assumptions appear idealized, insufficiently justified, or mathematically questionable, potentially weakening the theoretical contribution. Practical issues were also raised regarding computational overhead, memory scaling, and fairness of comparisons, as well as limited dataset diversity and missing analyses in the frequency domain.

Overall, while the paper presents a coherent and empirically motivated framework with promising results, reviewers remain divided on whether the conceptual and theoretical contributions are sufficiently strong and clearly differentiated from prior work.

**Reviewer Concerns:**

The reviewers have not provided the explicit feedback. Based on the interactions, and my evaluation, I think the following concerns remain unresolved,
1) Reviewer tWHh questions whether separable training is meaningfully different from existing frequency-band, residual, or modular training approaches (e.g., FEDformer, N-HiTS).

2) Theoretical validity. Reviewer 3iUt raises serious concerns about the assumptions used in Theorems 1 and 2, arguing that they may be unrealistic or mathematically unsound (e.g., assuming separable training achieves lower empirical loss, questionable expectation manipulations). These issues directly affect the credibility of the theoretical claims and are not fully resolved.

3) Reviewer 1DLx points out that, despite spectral bias being a central motivation, the experiments are confined to time-domain models. The absence of frequency-domain analysis or discussion limits the generality of the claims.

**Reviewer Scores:**

The reviewers have not provided the explicit feedback, and one Reviewer provides the strong negative point on this paper.

---

### Decision · Program_Chairs · 2026-01-26

Reject